# Harnessing whole human liver ex situ normothermic perfusion for preclinical AAV vector evaluation

Marti Cabanes-Creus[1], Sophia H. Y. Liao[1], Renina Gale Navarro[1], Maddison Knight[1], Deborah Nazareth[1], Ngee-Soon Lau[2,3], Mark Ly[2,3], Erhua Zhu[4], Ramon Roca-Pinilla [1], Ricardo Bugallo Delgado[5], Ana F. Vicente[5], Grober Baltazar[1], Adrian Westhaus [1], Jessica Merjane[1], Michael Crawford[2,3], Geoffrey W. McCaughan[2,6], Carmen Unzu [5], Gloria González-Aseguinolaza [5], Ian E. Alexander [4,7,8], Carlo Pulitano [2,3] & Leszek Lisowski [1,8,9] ✉

Developing clinically predictive model systems for evaluating gene transfer and gene editing technologies has become increasingly important in the era of personalized medicine. Liver-directed gene therapies present a unique challenge due to the complexity of the human liver. In this work, we describe the application of whole human liver explants in an ex situ normothermic perfusion system to evaluate a set of fourteen natural and bioengineered adeno-associated viral (AAV) vectors directly in human liver, in the presence and absence of neutralizing human sera. Under non-neutralizing conditions, the recently developed AAV variants, AAV-SYD12 and AAV-LK03, emerged as the most functional variants in terms of cellular uptake and transgene expression. However, when assessed in the presence of human plasma containing anti-AAV neutralizing antibodies (NAbs), vectors of human origin, specifically those derived from AAV2/AAV3b, were extensively neutralized, whereas AAV8-derived variants performed efficiently. This study demonstrates the potential of using normothermic liver perfusion as a model for early-stage testing of liver-focused gene therapies. The results offer preliminary insights that could help inform the development of more effective translational strategies.

As we venture further into the era of personalized medicine, the challenge of developing clinically predictive model systems for evaluating gene transfer and gene editing technologies presents a critical hurdle. The ideal model system should not only accurately mimic human physiological conditions and intricate tissue organization, but also reliably predict clinical outcomes. Within the world of liver-directed gene therapies, 2D and 3D in vitro models, including induced pluripotent stem cell (iPSC) derived organoids, murine models including xenograft models, and non-human primates have traditionally been employed to develop and validate novel advanced therapeutics. However, given the fact that the liver is a complex organ composed of multiple cell types that contribute to unique structural and functional organization, the complexity of the human liver has proven difficult to recapitulate. In response to this issue, particularly concerning adeno-associated viral (AAV) vector-based liver-targeted therapies, we herein present the development of a preclinical model based on whole human liver explants maintained in an ex situ normothermic (36 °C) perfusion system. This model provides a platform that allowed us, for the first time to the best of our knowledge, to assess the functionality of existing and additional recombinant adeno-associated viral (rAAV) vectors directly in the entirety of the human liver.

It is noteworthy that AAVs have recently attracted attention as effective and clinically proven gene therapy vectors. To date, five serotypes (AAV1, AAV2, AAV5, AAV9, and AAV-rh74) have earned regulatory approval for use in human patients[1], including one marketed product for the treatment of haemophilia (Hemgenix™, based on AAV5) and most recently another for Duchenne Muscular Dystrophy (based on AAVrh74). However, the current generation of AAVs remains suboptimal for the majority of clinical applications, often requiring the use of high vector doses[2]. Notably, prior to clinical evaluation, these AAV variants have been functionally tested in animal models, which while useful to evaluate safety and biodistribution, do not adequately recapitulate the intricate cellular dynamics and physiological responses of human livers.

To address this limitation, we turned our attention to normothermic human liver perfusion. The normothermic maintenance of the organ has been shown to reduce preservation-related graft injury compared to static cold storage in transplantable livers[3], thus offering a potentially superior model for evaluating systemic delivery and the complex interactions between AAV, the intravascular compartment, egress to the liver, and primary parenchymal and non-parenchymal human liver cells. Using the currently available protocols, livers are supplied with oxygen and nutrients at physiological temperature and pressures, maintaining conditions that support homeostasis, normal metabolic activity, and objective assessment of function in real-time[3]. More recently, ex situ normothermic perfusion has also been evaluated as a method to enable liver splitting before transplantation, aiming to help to address donor shortages by facilitating the transplant of one pediatric and one adult recipient from a single donor[4]. From the perspective of a preclinical model for the evaluation of advanced therapeutics, ex situ liver splitting also provides a promising model for evaluating gene therapy vectors and liver-directed biotherapeutics with a genetically matched control[4].

In this work, we used two whole human liver explants perfused with human blood to perform a functional evaluation of natural and bioengineered AAV vectors in the presence or absence of neutralizing antibodies. Specifically, we performed a next-generation sequencing (NGS) based parallel comparison of the transduction profile of a set of fourteen AAV variants (Supplementary Table 1) in whole human livers that were deemed unsuitable for transplantation. To provide preclinical context, we compared, in parallel, the same vector mix in other commonly used liver preclinical models, namely in the murine model, in xenografted mice engrafted with primary human or non-human primate hepatocytes, and in non-human primates.

Together, this work broadens the repertoire of preclinical models available for conducting liver-directed vector studies. In the future, this model could also be evaluated for studies aimed at developing novel bioengineered AAV variants and for disease-specific phenotype correction using gene addition and editing approaches. Concurrently this would allow the study of critical translational parameters such as therapeutic vector doses and potential vector-induced toxicity in perhaps what is the closest preclinical model of the human liver to date - the human liver itself.

## Results
### Study design
To enable functional evaluation of AAV variants in the context of the whole human liver, we adapted the Liver Assist perfusion system to allow for prolonged perfusion times of up to a week[5]. This system uses an open venous reservoir and incorporates two long-term oxygenators, a gas blender equipped with a pediatric flow regulator for ventilation control, and a flow-adjustable dialysis membrane for water-soluble toxin filtration and perfusate volume control (Fig. 1a).

To better recapitulate physiological conditions, we used a blood-based leukocyte-free perfusate, supplemented with essential nutritional supplements including amino acids, lipids, insulin, glucagon,

taurocholic acid, and methylprednisolone (Liver assist modifications, Methods)[4]. To minimize interference with AAVs that utilize heparan sulfate proteoglycan (HSPG) for cellular attachment, we supplemented the perfusate with enoxaparin (Clexane™), instead of standard heparin, as an anticoagulant (Supplementary Fig. 1). Before commencing the studies, we pre-screened numerous lots of fresh frozen plasma (FFP) for the presence of anti-AAV antibodies. This pre-screening allowed us to perfuse two livers differently; one with a perfusate containing AAV Nabs-positive plasma (creating neutralizing conditions) and the other with a perfusate containing plasma with low levels of NAbs (creating non-neutralizing conditions) (Fig. 1b).

To facilitate multiplexed vector comparison, we validated a collection of self-complementary AAV (scAAV) transgenes encoding a CAG-eGFP-BC-pA expression cassette containing a unique 44-mer barcode (BC) between the eGFP reporter and the poly(A) (Fig. 1b, Supplementary Figs. 2, 3). A unique barcoded construct was packaged into each of the fourteen selected AAV variants (Fig. 1b and Supplementary Tables 1 and 2). The selection included variants previously evaluated in clinical studies targeting the liver, such as AAV5, AAV8, and AAV-LK03, and also encompassed capsids used in clinical studies for organs other than the liver following systemic administration, such as AAV8 and AAV9. We also included several next-generation bioengineered capsids, such as AAV-SYD11 and AAV-SYD12[6], which showed superior efficiency at transduction of primary human hepatocytes in other pre-clinical models of the human liver. Each vector preparation was tittered individually, and combined at a 1:1 molar ratio, and the composition of the transgene mix was validated using NGS (Supplementary Fig. 4). To limit reperfusion injury and associated adverse effects, we incorporated the corticosteroid methylprednisolone into the liver explant perfusate. Building on a prior observation where prednisolone-treated animals showed a potential enhancement in vector genome and transgene expression[7], we evaluated its effect on the chosen AAV variants both in vitro and in a humanized mouse model. However, none of the AAVs used in our studies was shown to specifically benefit from prednisolone treatment at the studied concentrations (Supplementary Figs. 5, 6).

### Liver grafts, splitting and prolonged ex situ normothermic perfusion
Two whole human liver explants referred to as donor 1 (a 1.65 kg liver from a 78-year-old female) and donor 2 (a 2.19 kg liver from a 56-year-old male) were obtained from DonateLife, the centralized donation organization in Australia. These livers were unsuitable for transplantation and consented for research use. The first liver was excluded from transplantation due to biliary sepsis and cholecystostomy, while the second liver was donated after circulatory death (DCDD) and was deemed ineligible for transplantation due to the donor's age (>50), though it was otherwise healthy (Supplementary Table 2). The livers were procured as detailed in the Methods section (Liver origin and procurement). Following procurement, the livers were transported in static cold storage solution under ischemic conditions and remained in these conditions for 187 and 305 minutes, respectively, from the time of retrieval until the start of normothermic perfusion.

Both liver explants met viability-based inclusion criteria (outlined in the Methods section, Liver viability test) at the onset of the perfusion and again 24 hours later. Each liver was then split into two grafts: a left lateral sector graft (LLSG, consisting of segments 2 and 3) and an extended right graft (ERG, consisting of segments 1 and 4–8), following a recently described method[4]. These partial grafts were then perfused concurrently, each with its own modified Liver Assist perfusion system.

For donor 1, we achieved a portal flow rate of 1.6 L min$^{-1}$ and a pulsatile flow rate of 0.41 L min$^{-1}$ for the hepatic artery (Fig. 2a). After splitting the organ, the flow rates were adjusted in accordance with the

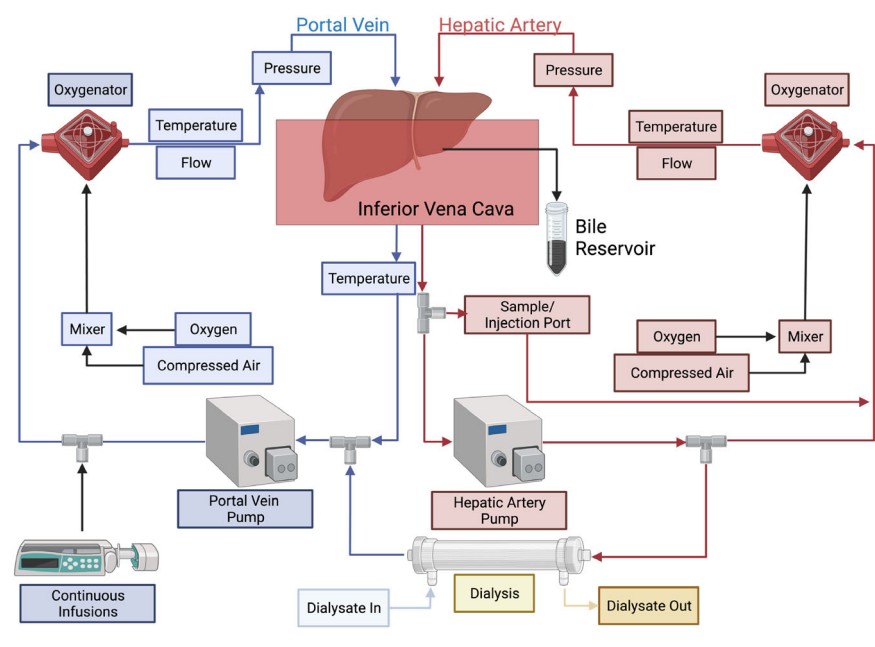

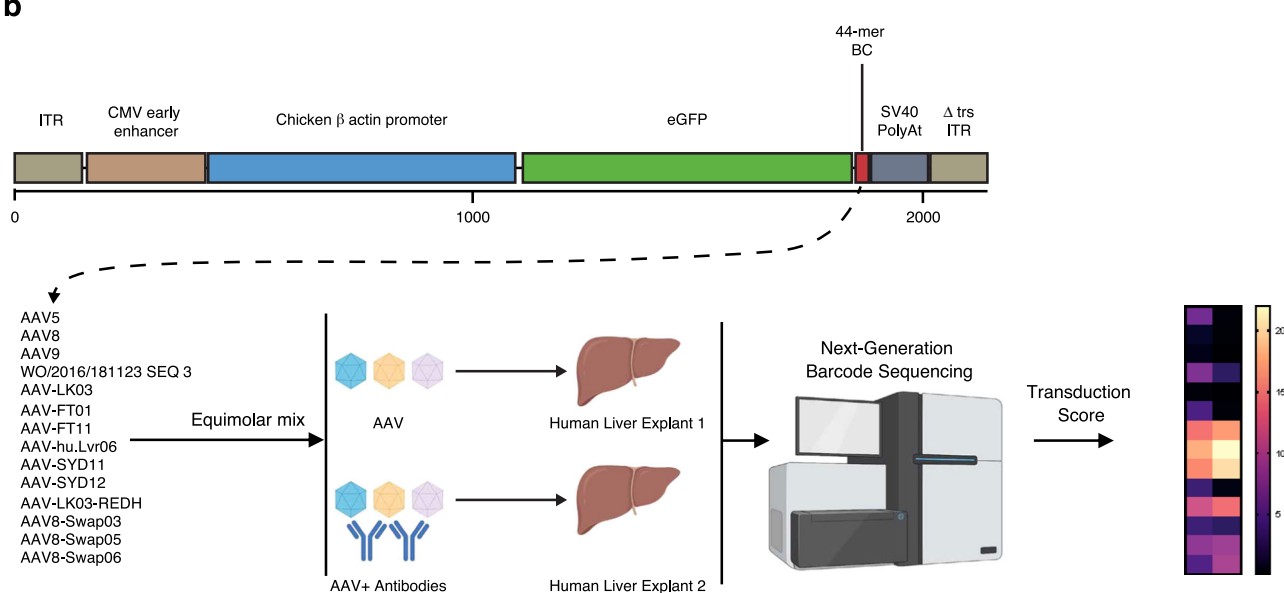

**Fig. 1 | Study design. a** Graphical flow chart of the liver perfusion machine. **b** Transgene composition and study design. CMV Cytomegalovirus, ITR Inverted Terminal Repeat.

vessel split; the main portal vein flow for the extended right graft and the main hepatic artery flow for the left lateral sector graft. This liver exhibited immediate lactate clearance (Fig. 2b) and progressively increasing bile production (Fig. 2c), which remained stable until day six before starting to decline. Core biopsies were taken at the start of the perfusion and every 24 hours after that (Fig. 2d, Supplementary Fig. 7), with no noticeable effect on the perfusate flow rate (Fig. 2a). Biopsies were scored *a posteriori* by a specialist pathologist for percentage of coagulative necrosis, hepatocyte detachment, and bile duct injury, which remained relatively stable (Supplementary Figs. 8–10). The degree of observed cell death was moderated and relatively constant across time (Supplementary Figs. 7–10), although signs of confluent necrosis could be seen as early as 48 hours post-perfusion (Supplementary Fig. 7).

We terminated the experiment at one week from the initiation of the perfusion. Liver function, as gauged by Factor V synthesis,

improved from an initial 13%, peaking at 29% for the LLSG 24 hours post-split and at 50% for the ERG 48 hours post-split (Supplementary Fig. 11). The liver transaminases release was high, as we recorded Alanine aminotransferase (ALT) at 522 U L$^{-1}$ at 24 hours post-split for the LLSG and 2,105 U L$^{-1}$ for the ERG at the same time point (Supplementary Fig. 12). We noted a similar trend with the acute cytokine levels in the perfusate, which began at a relatively low level (IL-6 = 78.8 ng L$^{-1}$, 4 hours post-perfusion), and spiked to over 10,000 ng L$^{-1}$ for both grafts by day five post-splitting (Supplementary Fig. 13).

In the case of donor 2, we observed similar trends in perfusate flows, lactate clearance, and bile production, but this liver remained stable for a slightly longer duration (Fig. 2e–g). Similarly to what was done in the case of liver from donor 1, we took core biopsies at the start of the perfusion and every 24 hours after (Fig. 2h, Supplementary Fig. 14), with no noticeable effect on perfusate flow (Fig. 2e). The

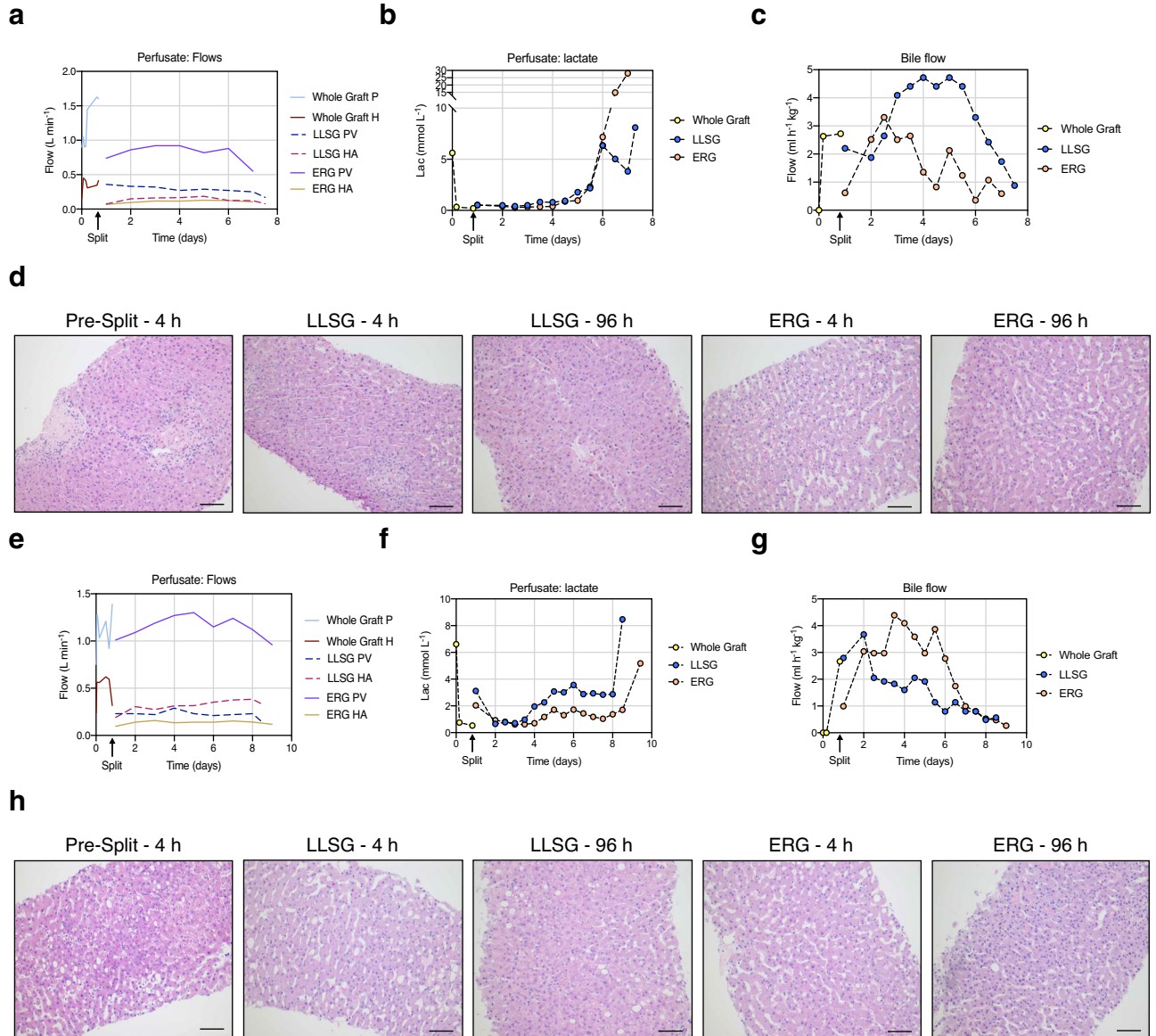

**Fig. 2 | Performance parameters during ex situ normothermic perfusion.**
**a**−**c** Course of perfusate flows, lactate concentration and bile flow throughout the perfusion of Donor 1. **d** Liver histology of representative core biopsies from Donor 1 at the indicated time points (n = 2 core biopsies per time point). Scale: 100 μm. **e**−**g** Course of perfusate flows, lactate concentration and bile flow throughout the perfusion of Donor 2. **h** Liver histology of representative core biopsies from Donor 2 at the indicated time points (n = 2 core biopsies per time point). Scale: 100 μm. PV Portal Vein, HA Hepatic Artery, LLSG Left Lateral Sector Graft, ERG Extended Right Graft.

increased functionality of this liver was reflected by the rise in Factor V synthesis from 14% at the beginning of perfusion, reaching a peak of 102% for the LLSG and 105% for the ERG five days post-split (Supplementary Fig. 15). The ALT levels peaked at 9762 U L⁻¹ at 4 hours post-split for the LLSG, and gradually decreased to 1663 U L⁻¹ on day 7.5 after splitting (Supplementary Fig. 16). This elevated ALT levels indicate severe hepatocyte damage during and after splitting. For the ERG, ALT levels rapidly increased, peaking at 6516 U L⁻¹, 4 days post-split, and then declining to 2273 U L⁻¹ on day 8 post-split (Supplementary Fig. 16). The acute cytokine IL-6 levels in the perfusate stayed relatively stable until day 7, at which point they rapidly increased to 1622 ng L⁻ for the left graft and at 15,744 ng L⁻ for the right graft (Supplementary Fig. 17). Overall, the results indicate that while both livers exhibited a degree of stability, signs of deterioration became evident at the five-day mark for the first liver and at the eight-day mark for the second liver. This suggests that the stability of the livers was compromised at these respective time points.

## Pre-screening of fresh frozen plasma for the presence of anti-AAV antibodies

In order to compare the efficiency of the chosen AAV variants (Supplementary Table 1) both with and without the interference of anti-AAV neutralizing antibodies (Nabs) (Fig. 1b), we first conducted a pre-screening of eighteen lots of human plasma. We followed a recently described ELISA procedure[8] to examine each plasma against the equimolar mixture of the fourteen AAV vectors. We classified the plasma samples based on their total reactivity (Supplementary Fig. 18).

Following preparation of the final plasma-containing perfusate, we examined the reactivity of each perfusate against each of the individual capsids present in the AAV mix. To do so, as outlined in the Methods section (ELISA measurement of anti-AAV IgG specific antibody titer in human serum), we evaluated the reactivity of each of the 14 capsid variants with the perfusate at dilutions of 1:25, 1:50, and 1:100. With the non-reactive perfusate used for donor 1 (Supplementary Fig. 19), we found all capsids to have an end titer of <1:25, excluding

AAV9, which showed minor reactivity at this dilution and is thus reported as presenting an end titer of 1:25. For the reactive perfusate used with donor 2, we found end titers to be 1:50 for the AAV3b-based variants (AAV-LK03, AAV-LK03-REDH, and AAV-SEQ-3) and for AAV-FT11, 1:25 for AAV-FT01, AAV-hu.Lvr06, AAV-SYD11, and AAV-SYD12, and non-reactive or <1:25 for AAV5, AAV8, AAV9, AAV8-Swap03, AAV8-Swap05, and AAV8-Swap06 (Supplementary Fig. 20).

## Functional evaluation of AAV vectors in the ex situ human liver perfused with non-neutralizing human plasma

With the appropriate human plasma samples identified, we proceeded to perfuse the first liver under non-neutralizing conditions. As schematically depicted in Fig. 3a, seven hours after starting the perfusion, and once the lactate levels had dropped from 5.62 mmol L$^{-1}$ to 0.33 mmol L$^{-1}$ (Fig. 2b), we injected the vector mix to the whole graft prior to liver splitting. We administered a total of $3.10 \times 10^{12}$ vector genomes (vg) directly into the portal vein. This dosage corresponds to $1.92 \times 10^{12}$ vg per kg of liver, which would equate to an estimated dose of $4.5 \times 10^{10}$ vg per kg of body weight. Considering the mix contained fourteen capsids, the estimated dose per capsid was $1.37 \times 10^{11}$ vg per kg of liver. Approximately fourteen hours post-AAV injection, we split the liver into the left lateral sector graft (LLSG) and the extended right graft (ERG), while maintaining uninterrupted arterial and portovenous perfusion (Methods, Liver assist modifications). Subsequently, we independently perfused and consistently monitored both grafts using separate Liver Assist machines. Notably, the LLSG continued to be perfused with the original perfusate containing AAVs, while the ERG received new AAV-free perfusate. This setup allowed us to investigate the kinetics of transduction at the fourteen-hour mark by comparing the AAV transduction profiles in both liver grafts (Fig. 3a).

We monitored vector clearance from the perfusate by analyzing plasma samples for the presence of AAVs at various times after vector infusion (Fig. 3b). Quantification of total AAV load indicated that starting from the initial estimated concentration of $1.55 \times 10^{9}$ vg mL$^{-1}$, the concentration of AAVs declined sharply to approximately 9% of these initial levels 24 hours post-injection (Fig. 3b). Subsequently, the AAV levels in the perfusate stabilized, suggesting that after the target organ absorbed the initial vector dosage, a noticeable fraction of around 10% of the vector mix persisted in circulation for the rest of the study.

To identify which of the AAV variants from the infused mix exhibited the lowest rates of clearance in the perfusate we performed NGS quantification of the unique barcoded region of the transgenes recovered from the perfusate, thereby demultiplexing the fraction of each AAV. In evaluating the percentage drop of each individual AAV over time in the perfusate, we distinguished three vector groups (Fig. 3b). A group of four variants - the HSPG de-targeted AAV-LK03-REDH, AAV-hu.Lvr06, AAV5, and AAV-FT01, were the primary contributors of the vectors present in the perfusate after the initial vector clearance (Fig. 3b). Following the initial drop, these variants persisted in the circulation at levels above 10% of the initial concentration. Another group, namely AAV9 and the HSPG-binders (AAV-LK03 and AAV-SEQ3), demonstrated the fastest clearance kinetics from the perfusate (Fig. 3b), with vector levels dropping to approximately 0.1% or less of the initial vector concentration within two days. All the remaining AAV variants exhibited an intermediate phenotype (Fig. 3b), with final concentrations varying between 0.1% and 1% of the individual initial levels.

We then turned our focus to the overall transduction profile of the vector mix in both liver grafts. To do this, we evaluated the total vector copy number per haploid cell in the DNA extracted from liver biopsies on days two to five post-injection. We found an increase in the average vector copy number in the left graft (LLSG) that continued to be perfused with the AAV-containing perfusate following the organ split (Fig. 3c). In contrast, the vector copy number in the right graft, which received AAV-free perfusate after the organ split, remained relatively stable at the tested timepoints, ranging from 1.5 to 1 vg per haploid genome (Fig. 3c).

To further analyze the transduction profile, we used NGS to examine the barcode composition in the DNA and RNA samples (which are indicative of cell entry performance and transgene expression, respectively) extracted from liver biopsy samples. We first calculated the percentage contribution of each vector to the total vector genomes in each graft, normalizing the data to the input pre-injection mix (Supplementary Fig. 4a). We then used unsupervised hierarchical clustering to group the AAV vectors with similar transduction profiles in the human liver, both at the DNA (vector uptake, Fig. 3d) and the RNA (functional transduction, Fig. 3e) levels.

As previously discussed, the right extended graft (ERG) provided a unique opportunity to study vector uptake during the first 14 hours post-injection. The AAV-SYDs and AAV-LK03 displayed rapid transduction kinetics, closely followed by the AAV3b-based AAV-SEQ3. In contrast, we found that AAV8, AAV9, AAV-FT1, and AAV-F11, AAV-hu.Lvr06, and AAV-LK03-REDH sat at the other end of the spectrum. These variants either did not efficiently transduce this liver or demonstrated slower kinetics of vector uptake (Fig. 3d, bottom panel). Looking at transduction kinetics, we found that the overall vector transduction performance remained stable throughout the experiment (Fig. 3d, bottom panel).

The transduction data we obtained from the left graft (LLSG) appeared more variable than the data for the right graft (refer to top panel of Fig. 3d). We believe this was likely a consequence of vector recirculation in the perfusate following organ split, which facilitated ongoing transduction during the course of the study. Specifically, in the LLSG, we detected increased relative transduction of some variants over time (AAV-SYD12, AAV-LK03, and AAV-SEQ3), while others exhibited decreased relative transduction over time (AAV-FT01, AAV-hu.Lvr06, AAV-LK03-REDH). Transduction for the remaining vectors remained stable.

Since these data present relative transduction, a decreasing contribution in the liver biopsies likely indicates either slower transduction kinetics or a decreasing amount of artificial signal stemming from the interstitial perfusate, rather than from vector uptake in hepatocytes. Indeed, the vectors showing a decreased relative transduction over time are the same one that stayed longer in the perfusate (Fig. 3c).

One interesting cluster of vectors comprised AAV-FT01, AAV-hu.Lvr06 and AAV-LK03-REDH. These vectors seemed to physically transduce the left graft but failed to transduce the right graft, which was exposed to the vectors for only the first fourteen hours. This likely indicates a slower transduction kinetics of these vectors (Fig. 3d).

Subsequently, we evaluated the barcode composition at the cDNA level in both grafts, serving as an indicator of each variant's efficiency at functionally transducing cells in this preclinical model of the human liver (Fig. 3e). Again, we employed unsupervised hierarchical clustering of the data to identify variants presenting similar functional transduction. Variants like AAV5, AAV-hu.Lvr06 and AAV-LK03-REDH did not functionally transduce the liver, despite the fact that these vectors were detected at the DNA level (Fig. 3e, Supplementary Fig. 21). Conversely, other variants like the AAV-SYDs, AAV-LK03, AAV-SEQ3, and AAV8-Swap05 (a variant harboring variable region (VR-I) from AAV2 and VRs VI to VIII from AAV7) exhibited similar functional transduction kinetic, with levels of reads recovered from cDNA closely matching the levels recovered from DNA (Supplementary Fig. 21). The AAV8-Swap06 variant displayed faster functional transduction kinetics, evident by a higher relative percentage of cDNA reads (~2×) compared to DNA reads at all the studied time points (Supplementary Fig. 21), a phenomenon we had previously observed in mice with humanized livers[6].

Lastly, we conducted microscopic analysis of the transduced liver using immunofluorescence (Fig. 3f–k) as well as hematoxylin and eosin staining (H&E, Fig. 2d, Supplementary Fig. 7). We could readily detect

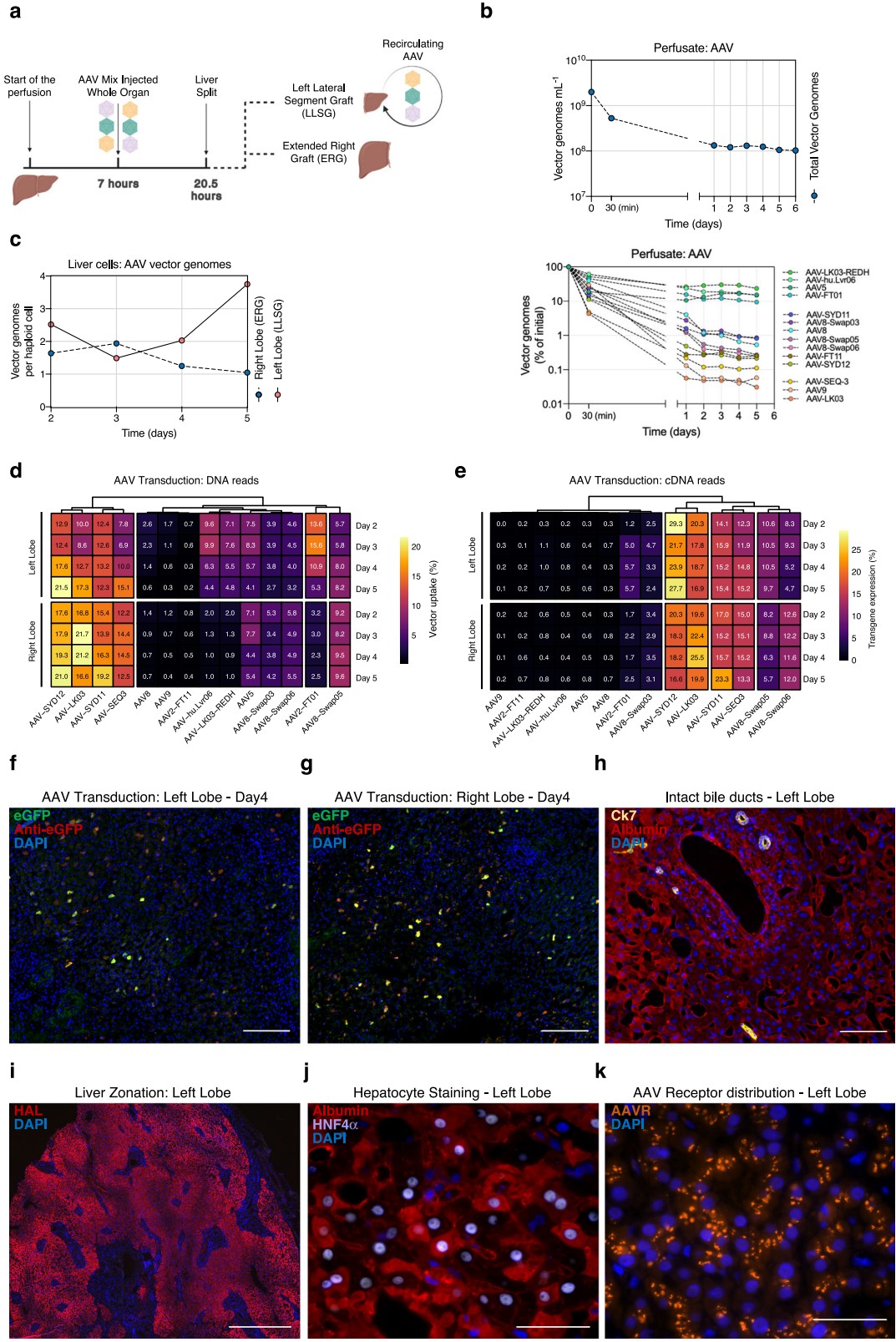

eGFP signal arising from collective vector transduction at day 4 post-injection in both grafts (Fig. 3f, g, Supplementary Figs. 22 and 23). Notably, there was an absence of distinctive vector zonation, which could be attributed to the diversity of capsids present in the vector mix used in the study. Intrahepatic bile ducts appeared well-preserved (Fig. 3h), as was liver zonation, demonstrated by staining of the portal marker histidine ammonia-lyase (HAL, Fig. 3i). We confirmed liver parenchymal and non-parenchymal integrity through staining for albumin and the hepatocyte nuclear factor 4 (HNF4A), which stains for human hepatocytes nuclei (Fig. 3j). Finally, to gain insights into the mechanisms of cell-AAV interaction, we analyzed the distribution of the AAV-Receptor (AAVR) in the human liver explant (Fig. 3k). AAVR could be detected in human hepatocytes, although the localization of appeared to be primarily intracellular (Fig. 3k).

**Fig. 3 | Functional evaluation of AAV vectors in the ex situ human liver perfused with non-neutralizing human plasma. a** Graphical representation of the study. **b** Concentration of total (top panel) and de-multiplexed AAV vector genomes in the perfusate (bottom panel). Concentration of individual vectors is expressed as the estimated percentage of the initial concentration as studied with NGS. Vectors are ranked from most to least abundant (top to bottom). **c** Total vector genomes per haploid cell found in biopsies from both liver grafts extracted at the indicated perfusion times. **d** Percentage of NGS reads mapped to each barcoded AAV capsid variant. The transgene DNA, indicating vector uptake, was extracted from biopsies taken from both grafts at the indicated time points. **e** Similar analysis performed on transgenes recovered from RNA, which indicate functional transduction. Percentages are normalized to the pre-injection mix. **f** Immunofluorescence analysis of the net eGFP signal from collective AAV transduction in the left graft. **g** Similar immunofluorescence image for the right graft. The vector-encoded eGFP was also counterstained with an anti-eGFP antibody (red). Blue: DAPI (nuclei). Scale = 100 μm. **h** Immunofluorescence analysis of the bile ducts and hepatocytes of the left graft. Yellow: Cytokeratin 7 (bile ducts); red: albumin; blue: DAPI (nuclei). Scale: 50 μm. **i** Immunofluorescence analysis of liver zonation. Red: Histidine ammonia-lyase (HAL); blue: DAPI (nuclei). Scale: 500 μm. **j** Immunofluorescence analysis of liver integrity. Red: albumin; Purple: Hepatocyte nuclear factor 4 alpha (HNF4α); blue: DAPI (non-hepatocyte nuclei). Scale: 20 μm. **k** Immunofluorescence analysis of AAV Receptor (AAVR) in the human liver. Orange: AAVR; Blue: DAPI (nuclei). Scale: 100 μm.

## Functional evaluation of AAV vectors in the ex situ human liver perfused with neutralizing human plasma

In the second experiment, we perfused the liver using plasma containing antibodies reactive against a subset of capsids present in the vector mix (Supplementary Fig. 20). Unlike the first study detailed earlier, we only injected the vector mix into the left graft (Fig. 4a), post-liver split, once the lactate level had reduced from 3.12 mmol L$^{-1}$ to 0.65 mmol L$^{-1}$ (Fig. 2f). The right graft did not receive any AAV and acted as a donor-matched, untreated control. In addition, to counter the presence of neutralizing antibodies for some of the capsids, we doubled the dose of AAV per kg of liver relative to the first liver. Specifically, we injected a total of $2.57 \times 10^{12}$ vector genomes into the left portal vein. Given that the LLSG weighted 0.67 kg, this corresponded to a total dose of $3.83 \times 10^{12}$ vg per kg of liver, or $2.74 \times 10^{11}$ vg per kg of liver for individual variants present in the mix.

As in the case of donor 1, we initially measured vector clearance from the AAV-containing perfusate. In stark contrast to our observations under non-neutralizing conditions, where vector genomes stabilized at around $1 \times 10^{8}$ vg mL$^{-1}$ (Fig. 3b), we observed a rapid disappearance of vector genomes from the perfusate to below $1 \times 10^{6}$ vg mL$^{-1}$ by day 4 (Fig. 4b), equivalent to ~0.06% of the estimated initial concentration of $1.29 \times 10^{9}$ vg mL$^{-1}$. Next, we employed NGS to investigate the clearance kinetics of individual variants from the perfusate. Given the time-course of vector clearance (Fig. 4b), we focused on the changes in the composition of the AAV mix present in the perfusate during the first three days of the study. We identified two distinct vector populations; i) a group of five variants (AAV9, AAV5, AAV8, AAV8-Swap03 and AAV8-Swap06) that exhibited slow clearance kinetics, with each variant present at levels above 1% of the initial vector composition on day 3 post-injection (Fig. 4b, lower panel) and ii) a group consisting all remaining variants that were rapidly cleared from the system, with levels ranging from 0.001% to 0.01% of their corresponding initial levels, one-log lower than the lowest levels previously observed under non-neutralizing conditions (Fig. 3b). Notably, there appeared to be a correlation between high plasma reactivity and faster vector clearance kinetics from the perfusate, as all vectors present on day 3 at <1% of the initial concentration displayed an end Nab titer of 1:25 or >1:50 (Supplementary Fig. 20). This could suggest that the anti-AAV neutralizing antibodies present in the perfusate were directly responsible for vector removal from the circulation, most likely through opsonization followed by residential macrophage clearance. In fact, we found a statistically significant Spearman negative correlation between plasma reactivity and perfusate clearance (Day 1: r = −0.9165, p < $2.2 \times 10^{-16}$; Day 2: r = −0.8725, p < $2.2 \times 10^{-16}$; Day 3: r = −0.8901, p < $2.2 \times 10^{-16}$), in support of this hypothesis.

Analysis of the vector copy number in the biopsy samples taken between day 2 and day 6 post vector infusion indicated an average of ~1.5 vector genomes per haploid cell up to day six of the study, followed by a small yet noticeable decrease to around 0.5 vector genomes per haploid cell at the conclusion of the study on day 7 and 8 (Fig. 4c). Next, we used NGS to examine the transduction profile of individual AAVs in the left graft. Figure 4d shows the hierarchical clustering of vector uptake (at the DNA level) contribution after normalization to the pre-injection input mix (Supplementary Fig. 4b). In marked contrast with the non-neutralizing conditions for donor 1 (Fig. 3d), we did not detect AAV-LK03 uptake for this donor, and only a relatively low number of reads mapped to AAV-SYD12 (Fig. 4d). Generally, the presence of neutralizing antibodies for specific capsids effectively obstructed liver uptake of those variants (Fig. 4d). The only capsids we could readily detect in this liver were those that did not react with this batch of human plasma, namely AAV5, AAV8, AAV9, AAV8-Swap03 and AAV8-Swap06 (Supplementary Fig. 20). AAV8-Swap05, which displayed the slowest kinetics of clearance from perfusate among the group of reactive variants (Fig. 4b, lower panel), also exhibited a slightly lower antibody reactivity at 1:25 (Supplementary Fig. 20). Interestingly, AAV8-Swap03, which contains variable regions VI to VIII from AAV7[6], functioned particularly well under these neutralizing conditions. In terms of functional transduction, or cDNA reads, we noticed a trend similar to that observed for vector uptake. The sole exception was AAV5, which functioned relatively less efficiently than the other variants, especially considering its relatively higher vector uptake (Fig. 4e, Supplementary Fig. 21).

In the subsequent analysis of this liver using immunofluorescence (Fig. 4f–k) and H&E staining (Fig. 2h, Supplementary Fig. 7), we observed a similar pattern to that of donor 1. On day 4 post-injection, we could readily detect the net eGFP signal from collective transduction by the vectors present in the mix (Fig. 4f). Taking into account the fact that the vector particles reactive with anti-AAV neutralizing antibodies (Nabs) present in the plasma were eliminated from this closed circulation (since they were undetectable in the DNA purified from biopsy samples as well as in the perfusate) and considering that the dialysis filters used had a cut-off of 60 kDa, which is substantially below 4751 kDa estimated size of full AAV8 particles[9], thereby precluding their filtration, we explored the possibility that Kupffer cells (the resident macrophages in the liver) were responsible for this active vector clearance. As shown in Fig. 4g consistent with the fact that this was an entire human liver organ, we could detect a large number of Kupffer cells (CD68 + ) in the liver. Notably, just as described for the liver from donor 1, histological analyzes confirmed that the intra-hepatic bile ducts were well-preserved (Fig. 4h), as was metabolic zonation (Fig. 4i), parenchymal integrity (Fig. 4j) and wide lobular expression of the AAV Receptor (Fig. 4k).

Finally, to place the results observed in the human liver explant in the context of other established preclinical models of human liver, we studied the relative performance of the same set of fourteen vectors (Supplementary Table 1) in the murine liver, the xenograft mouse models engrafted with human and non-human primate hepatocytes, as well as in vivo in a non-human primate (Supplementary Table 4). The results are detailed in Supplementary Figs. 24−29 and in the Supplementary Note. Briefly, AAV-LK03 and AAV-SEQ3 demonstrated a strong ability to perform well in the non-human primate model (Supplementary Fig. 26), as well as in human liver explants when

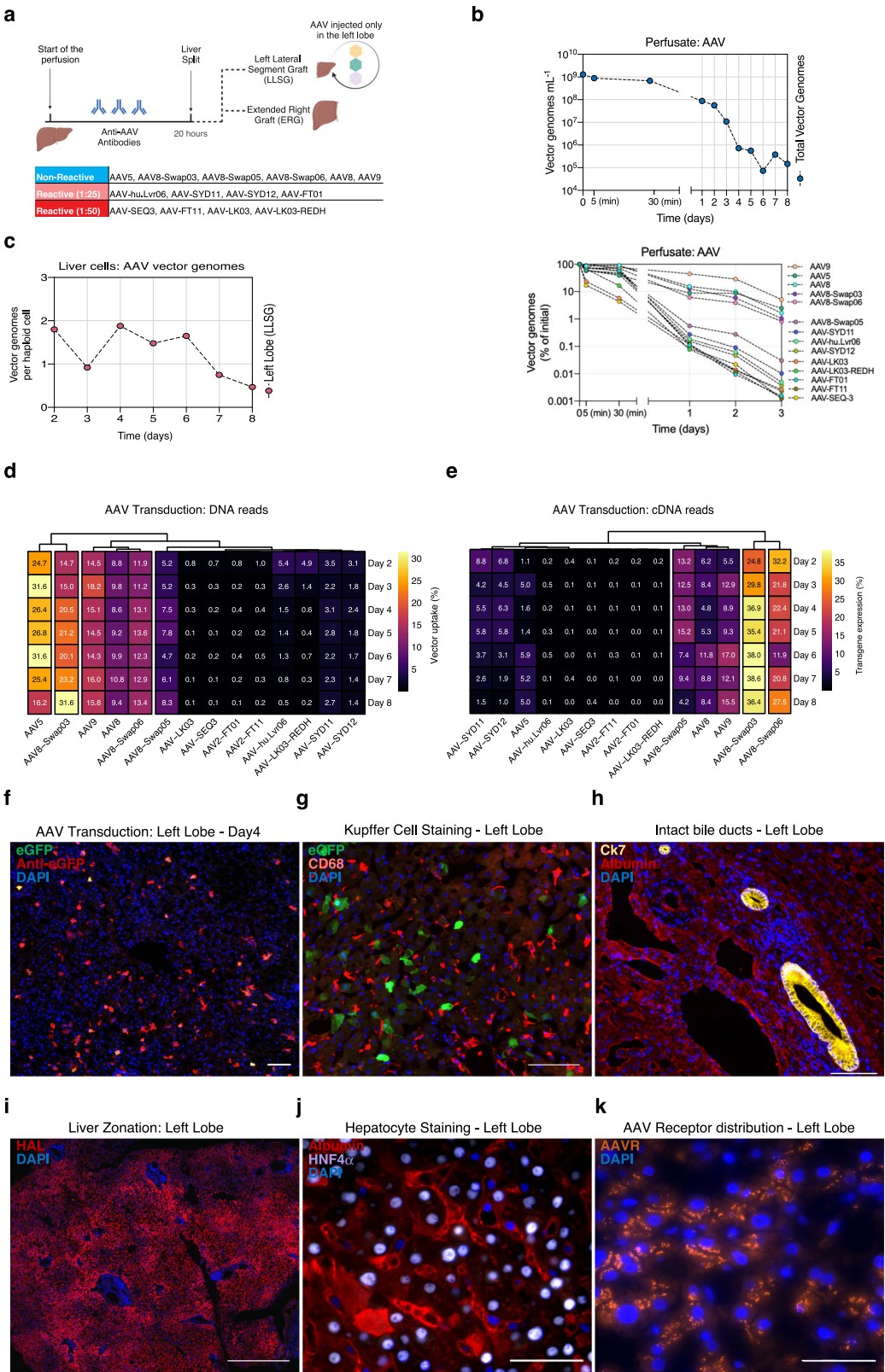

neutralizing antibodies were not present (Fig. 3), suggesting these vectors have a high potential for effective gene delivery in liver tissues. Additionally, we found AAV-SYD11 and AAV-SYD12 be effective in FRG mice engrafted with human and non-human primate hepatocytes (Supplementary Figs. 27, 28). However, these two vectors performed poorly in the non-human primate (Supplementary Fig. 26). Despite AAV5's consistent success in entering liver cells across various models,

it failed to show a corresponding level of success in expressing genes once inside the cells, highlighting a challenge in achieving successful gene expression post-cell entry.

The variation in vector performance across different models underscores the complexity of preclinical vector assessment and the need for comprehensive evaluation across multiple systems to ensure translational relevance of these findings.

**Fig. 4 | Functional evaluation of AAV vectors in the ex situ human liver perfused with neutralizing human plasma. a** Graphical representation of the study. **b** Concentration of total (top panel) and de-multiplexed AAV vector genomes in the perfusate (bottom panel). Concentration of individual vectors is expressed as the estimated percentage of the initial concentration as studied with NGS. Vectors are ranked from most to least abundant (top to bottom). **c** Total vector genomes per haploid cell found in biopsies the left graft. **d** Percentage of NGS reads mapped to each barcoded AAV capsid variant. The transgene DNA, indicating vector uptake, was extracted from biopsies taken from both grafts at the indicated time points. **e** Similar analysis performed on transgenes recovered from RNA, which indicate functional transduction. Percentages are normalized to the pre-injection mix. **f** Immunofluorescence analysis of the net eGFP signal from collective AAV

transduction in the left graft. The vector-encoded eGFP was also counterstained with an anti-eGFP antibody (red). Blue: DAPI (nuclei). Scale = 100 μm. **g** Immunofluorescence analysis of the presence of Kupffer cells in the left graft. Green: vector-encoded eGFP; Red: CD68 (Human macrophage marker); blue: DAPI (nuclei). Scale 100 μm. **h** Immunofluorescence analysis of the bile ducts and hepatocytes of the left graft. Yellow: Cytokeratin 7 (bile ducts); red: albumin; blue: DAPI (nuclei). Scale: 50 μm. **i** Immunofluorescence analysis of liver zonation. Red: Histidine ammonia-lyase (HAL); blue: DAPI (nuclei). Scale: 500 μm. **j** Immunofluorescence analysis of liver integrity. Red: albumin; Purple: Hepatocyte nuclear factor 4 alpha (HNF4α); blue: DAPI (non-hepatocyte nuclei). Scale: 20 μm. **k** Immunofluorescence analysis of AAV Receptor (AAVR) in the human liver. Orange: AAVR; Blue: DAPI (nuclei). Scale: 100 μm.

## Discussion

In 1976, a British statistician George Box[10] made a famous statement: "Remember that all models are wrong; the practical question is how wrong do they have to be to not be useful". This has become an enduring reminder that we must constantly assess the efficacy of the models we employ in research. In this study, we pioneered the use of an ex situ whole human liver model that, while not free of limitations, provides a distinct platform to explore the intricate workings of AAV vector performance. The model, involving a human organ perfused with human blood and maintained at body temperature, offers a strikingly accurate approximation of the human physiological setting. One could argue that there are few alternatives that bring us closer to the human setting, short of an actual clinical trial. This signifies the extraordinary potential of this model in accelerating progress in the field of liver-mediated gene therapies – both viral and potentially non-viral - by enabling investigations that closely mimic the clinical scenarios where gene therapies will ultimately be deployed. Given the complexity and potential ethical questions surrounding this model, it is imperative to emphasize that the primary intent for donated livers is transplantation, aiming to provide direct lifesaving benefits to recipients in need. However, in instances like those described here, where livers were unsuitable for transplantation, redirecting these organs for research use offers an alternative ethical use of those organs, contingent upon the presence of an approved human research protocol and an appropriate informed consent.

As evidenced in our study, ex situ human liver explants can be maintained in a viable state over an extended period, thereby providing a sufficiently long experimental window to facilitate a broad range of studies, including the development and evaluation of viral vectors. In the perfusions detailed here, liver function remained stable up to the fifth day, as demonstrated by the steady perfusate flow, consistently low lactate concentration in the perfusate, and sustained bile production (Fig. 2). Nonetheless, a posteriori analysis of the samples revealed a substantial release of alanine aminotransferase (ALT) (Supplementary Figs. 12 and 16). Considering the fact that these increases were already evident prior to vector injection and in samples collected from the control untreated right graft of donor 2, which was not exposed to the AAV vector (Fig. 4a), it is reasonable to attribute the elevated ALT to reperfusion injury. Reperfusion injury is a significant challenge in liver transplantation[11] and, in this instance, was likely unrelated to the vector transduction process. In our study, we incorporated the use of methylprednisolone to limit reperfusion injury and associated adverse effects. Nonetheless, future research is necessary to pinpoint the most effective treatment regimen.

While the liver explant system offers a valuable perspective for examining antibody neutralization in an environment containing liver-resident immune cells, it is limited by the exclusion of circulating immune cells and other systemic factors involved in antibody-mediated antigen clearance. These limitations may affect the system's ability to fully replicate the complex interactions and clearance mechanisms observed in the context of a whole-body. Intriguingly, we noted a rapid and almost complete neutralization of AAV variants that

displayed reactivity to neutralizing antibodies (NAbs) on end-point titers of 1:50 and above. This led to rapid clearance from the perfusate (Fig. 4b), an effect we hypothesize to be mediated by liver-resident Kupffer cells. It is noteworthy that this phenomenon would be challenging to replicate in any existing preclinical liver model. We observed minor reactivity of the overall non-reactive perfusate (from donor 1) to AAV9 (Supplementary Fig. 19), which exhibited rapid clearance from the perfusate under these conditions (Fig. 3b, lower panel). Consequently, it is reasonable to hypothesize that AAV9's modest performance in donor 1 (Fig. 3d, e) could be due to antibody neutralization. Considering the fast clearance kinetics observed when the vector mix was directly injected into the portal vein, systemic administration of vectors may prove to be even more challenging. Interestingly, and perhaps predictably, given our use of plasma from human donors, who are more likely to possess antibodies against viruses infecting humans, the vectors that performed best under non-neutralizing conditions were also those most heavily neutralized. Conversely, the AAV vectors that managed to evade this specific human plasma performed relatively poorly under non-neutralizing conditions. This observation hints at a potential trade-off between function and antibody neutralization. These findings also reinforce the hypothesis that under neutralizing conditions, strategies aimed at circumventing anti-AAV antibodies, such as the employment of endopeptidases[12], could prove critically important in enhancing the potential for patients with detectable anti-AAV NAbs to benefit from innovative gene therapies. While we observed a notable correlation between the presence of neutralizing antibodies and the obstruction of liver uptake of specific capsids, we must acknowledge the potential impact of specific donor-related factors on our findings. Specifically, one of the livers was excluded from transplantation due to biliary sepsis, and the second liver was procured from a donor after circulatory death (DCD). Hence, the impact of both the infection and the warm ischemia may be relevant and could affect vector performance, adding a level of complexity to the interpretation of the results of our study. Thus, the observed elevation of IL-6 and ALT in donor 1 presents a potential confounding factor in the assessment of transduction efficacy of the capsids. Our analysis leads us to hypothesize that the primary determinant of the observed discrepancies between donor 1 and donor 2 is related to the presence of neutralizing antibodies in the case of donor 2. These antibodies likely exert a more pronounced effect on the transduction efficiency, overshadowing the potential impact of the elevated IL-6. Nonetheless, the role of IL-6 as a contributor to the variability in transduction cannot be entirely discounted and warrants further investigation. Furthermore, with only two livers utilized, our sample size is too small to account for the heterogeneity inherent in human liver physiology. Factors such as varying genetic backgrounds between both donors, as well as age, sex, and disease states, could potentially influence individual capsid performance. These variables underscore the significance of the limitation given the complex nature of liver functions and the variable response to normothermic machine perfusion (NMP) based on individual liver pathology and history. These constraints are critically highlighted in

the context of our results and should be carefully considered[13]. To achieve broader applicability and address these limitations, further research with a larger and more diverse range of liver organs is necessary to validate and generalize our findings.

A second powerful feature of the human liver explant model is the potential it offers for aligning vector functionality with historical clinical data. In our study, we administered a relatively low vector dose to both livers, a quantity significantly smaller than those utilized in some clinical trials. For donor 1, the injection of the vector mix into the entire organ allowed for an estimation of the equivalent vector dose per kg of body weight, which amounted to approximately $4.5 \times 10^{10}$ vg per kg of body mass, based on the known body mass of the donor of liver 1. As the mix contained 14 AAV variants, the calculated dose per capsid was only $3.2 \times 10^9$ vg per kg of body weight. This represents a figure nearly 20,000 times lower than the dose administered in Roctavian, the recently approved AAV5-based product for Hemophilia A[14–18]. Our inability to detect transgene expression at the studied timepoints for AAV5 implies that a higher dose might be necessary for this specific serotype (Fig. 3e). However, the detection of eGFP expression under both neutralizing and non-neutralizing conditions, attributable to other liver-tropic capsids present in the mix, suggests that lower clinical vector doses may be sufficient to achieve clinical benefits if AAV variants that transduce human primary hepatocytes with high efficiency are used.

Finally, the ex situ human liver explant model introduces an exciting avenue for the development of distinct AAV vectors through directed evolution approaches. Its unique feature of maintaining native liver structure and human extracellular matrices makes it an ideal platform for refining strategies such as the HSPG de-targeting described previously in the humanized FRG model[19].

All in all, we believe the human liver explant is among the most biologically and clinically predictive preclinical models available today, since it provides the valuable opportunity to perform vector development and evaluation in the context of the whole organ with intact organ architecture and zonation, in the presence of human blood. Further work needs to be conducted to understand convergences and divergences between the available preclinical models and how these relate to the accumulated clinical data. It is essential to highlight that this model is complementary to other in vivo models, such as the FRG or NHP, which continue to offer invaluable insights into vector biodistribution and AAV-mediated cellular toxicity.

In conclusion, while recognizing the inherent challenges, including the limited availability of whole human livers, procedural costs, genetic background variations, and the effects of perfusion-induced cellular stress on AAV transduction, we view this work as a compelling initial proof of concept, paving the way for future studies that aim to address and overcome the mentioned challenges. In this sense, we believe the whole human liver explant model offers an exceptional opportunity to study AAV vector function in a model that closely recapitulates human conditions. This model, with time, has the potential to lead to the development of the next-generation human liver-targeted (and human liver de-targeted) AAV variants, starting a new era of potentially successful gene therapy liver-directed clinical trials.

## Methods
### Ethical statement
The human livers used throughout this study were accepted in accordance with a study protocol approved by the Sydney Local Health District Ethics Review Committee (X18-0523 & 2019/ETH08964). All the murine experimental procedures and care were approved by the joint Children's Medical Research Institute (CMRI) and The Children's Hospital at Westmead Animal Care and Ethics Committee. Non-human primate procedures were approved by the ethical committee for animal testing of the University of Navarra and

by the Department of Health of the government of Navarra (Comité de Etica para la Experimentación Animal code: 038/15) and performed according to the guidelines from the institutional ethics commission.

### AAV transgene constructs
All the vectors used in the study contain AAV2 ITR sequences. The 44-mer long barcodes were cloned using standard molecular biological techniques into a self-complementary, CAG plasmid driving eGFP expression (Addgene #8327941). The sequences of such barcodes can be found in Supplementary Table 2. The barcodes were cloned between the fluorophore and the Simian virus 40 (SV40) poly-adenylation as indicated in Fig. 1b.

### AAV vector packaging and viral production
AAV constructs were packaged into AAV capsids using human embryonic kidney (HEK) 293 T cells and a helper-virus–free system[20]. All AAV capsids used in this study were produced using poly-ethylenimine (PEI) transfection of the self-complementary CAG-eGFP-barcode transgene cassette, as well as adenovirus and Rep2-Cap helper plasmids[21]. Briefly, 22.5 μg of the Adenovirus 5 helper plasmid, 7.5 μg of the Rep2-Cap plasmid, and 7.5 μg of the self-complementary CAG-eGFP-barcode transgene cassette were transfected per 15 cm plate. Briefly, each capsid serotype was co-transfected individually with the corresponding unique barcoded transgene. The resulting cell lysates were tittered individually and subsequently mixed at equimolar ratio. Specifically, the transfected cells were harvested 72 hours post-transfection and subsequently centrifuged. The cell pellets obtained from the initial centrifugation were resuspended in 400 mL of resuspension buffer (PBS, 10 mM Tris-HCl pH 8.5, 2 mM MgCl$_2$) per 15 cm plate, and subjected to three cycles of freeze-thaw using dry ice/ethanol and a 37 °C water bath. Following the third cycle, Benzonase (from EMD Chemicals, Merck, #1.01695.0002) was added to achieve a final concentration of 200 U/mL, and the mixture was incubated for an hour at 37 °C. The cell suspension was then centrifuged to remove cell debris, and the clear supernatant was transferred into a new 50 mL Falcon tube. To this, 1/4 volume of 5 M NaCl and 10% Sodium deoxycholate to a final concentration of 0.5% were added. The suspension containing AAV particles was again incubated for 30 min at 37 °C and subsequently spun at $5250 \times g$ at 4 °C. The supernatant was then moved into a new tube and tittered. Each capsid/transgene was then mixed equimolarly at this stage. The vector mix was then purified using Cesium Chloride (CsCl, Ultrapure optical grade, from Life Technologies, catalog number 15507-023) gradient ultracentrifugation following the previously published protocol[20]. Specifically, this process involved adding 12 mL of 1.3 g/mL CsCl solution in Dulbecco's Phosphate Buffered Saline (from Sigma Aldrich #D8537) to an Ultra-Clear Centrifuge Tube (Beckman Coulter, #344058) and layering 5 mL of 1.5 g/mL CsCl solution to form a clear interface. The 20 mL sample of cell pellet solution was carefully layered on top, and the gradients were spun in a SW32 Ti Rotor at 25 K ($106,800 \times g$) at 20 °C for 24 hours. The vector fractions from the distinct CsCl density interfaces were collected using a 10 mL syringe with an 18 G needle, combined, and adjusted to a CsCl final density of 1.37 g/mL for a second round of centrifugation. This second centrifugation was performed in a SW41 Ti rotor at 38 K ($247,600 \times g$) for 24 hours at 20 °C. The clear band corresponding to the concentrated AAV vectors was subsequently extracted with a syringe. The combined AAV solution was then loaded into a dialysis cassette (10,000 MWCO, 3.0 mL capacity, Thermo Fisher, #87730). The vector preparation was dialyzed twice in 2 L of PBS at 4 °C with stirring for 24 hours. An additional dialysis was conducted in 5% Sorbitol in PBS for 3 hours at 4 °C. The AAV particles were then retrieved from the dialysis cassette and passed through a 0.22 μm syringe filter (Merck Millipore, # SLGP033RS). The final 3 mL of vector solution was concentrated to 1 mL using an Amicon Ultra-15 Centrifugal Unit (100 kDa NMWL, from Merck Millipore, # UFC910024).

## AAV titration

AAV titration was performed via digital droplet PCR (ddPCR, Bio-Rad, Berkeley, CA, USA) using EvaGreen supermix (Bio-Rad, catalog no. 1864034) and following the manufacturer's instructions. To detect AAV genomes on vectors, GFP primers were used (GFP-F: 5′- TCAA GATCCGCCACAACATC; GFP-R: 5′- TTCTCGTTGGGGTCTTTGCT).

## Barcode Amplification, NGS, and distribution analysis

To amplify the transgene region containing the 44-mer barcodes, BC-F (5′- GAGTTCGTGACCGCCG) and BC-R ('5- ATTGCAGCTTATAATGG TTACAAATAAAGC) were used. NGS library preparations and sequencing using 2 × 150 paired-end configurations were performed by Azenta (Suzhou, China) using an Illumina HiSeq instrument. A workflow was written in Snakemake (5.6)[42] to process reads and count barcodes. Paired reads were merged using BBMerge and then filtered for reads of the expected length in a second pass through BBDuk, both from BBTools 38.68 (https://sourceforge.net/projects/bbmap/). The merged, filtered fastq files were passed to a Python (3.7) script[22] that identified barcodes corresponding to AAV variants. NGS reads from the DNA and cDNA populations were normalized to the reads from the pre-injection, vector mix.

## Cell culture, vector transduction, and heparin and enoxaparin (Clexane) competition assay

HuH-7 cells were kindly provided by Dr Jerome Laurence (The University of Sydney). HEK293T cells were obtained from ATCC (Cat#CRL-3216). All cells were tested for mycoplasma and were mycoplasma-free. Cells were cultured as described previously[23] with no modifications. Specifically, HEK293 cells were cultured in Dulbecco's Modified Eagle's Medium (DMEM) supplemented with 10% fetal bovine serum (FBS), 1% penicillin/streptomycin (P/S), and incubated in humidified 37 °C incubator with 5% CO2. HuH-7 cells were cultured under the same conditions, with the addition of 1% non-essential amino acids (NEA) in the medium. For the heparin competition assay (Supplementary Fig. 1), cells were seeded at $10^5$ per well into 24-well plates at day 0 and transduced at the indicated vector genome/cell. When indicated, heparin sodium salt (Sigma, H3149-50KU, lot no. SLBW2119) or Clexane was supplemented at 50 µg/mL or at 200 µg/mL. After 72 h, the cells were harvested using TrypLE express and analyzed for GFP using BD LSRFortessa cell analyzer. The data were analyzed using FlowJo 7.6.1.

## RNA stability test of the barcoded constructs

To minimize a possible bias in transgene RNA stability introduced by the longer barcodes, 25 individual barcoded constructs were packaged in AAV2 and AAV-DJ, and those constructs leading to either over-expression or under expression of the transgene when compared to the mean transduction values were discarded (Supplementary Fig. 2). Briefly, the 25 barcoded transgenes were mixed at 1:1 molar ratio and co-transfected into HEK293T as described below. After vector purification, HuH-7 cells were transduced as described previously with no modifications, with a multiplicity of infection of 50, 500, and 5000 vector genomes per cell. Three days after transduction, cells were harvested and the barcoded region was PCR-amplified from the vector preparation, DNA and RNA extracted from the cells, as described below. Fifteen barcodes (Supplementary Table 3) showing low bias were further characterized in vitro and in vivo (Supplementary Fig. 3).

## Analysis of the in vitro influence of methylprednisolone on the transduction efficiency of AAV vectors

HuH-7 cells were seeded at $10^5$ per well into 24-well plates at day 0 with and without the presence of methylprednisolone (Supplementary Fig. 5). Cells were transduced with the barcoded vector mix containing the 14 studied capsids at day 1 (10,000 vg/cell, n = 6 per condition). At day 2, fresh media and methylprednisolone were added, and cells were subsequently harvested at day 3 using TrypLE express. Approximately half of the cells were analyzed for GFP using BD LSRFortessa cell analyzer. The data were analyzed using FlowJo 7.6.1 (Supplementary Fig. 5). DNA and RNA were extracted from the remaining cells. Vector copy number was analyzed as described below. The composition of barcodes at the entry and expression level were analyzed with NGS as described above (Supplementary Fig. 5).

## Analysis of the in vivo influence of methylprednisolone on the transduction efficiency of AAV vectors

To study the potential effect of methylprednisolone on transduction in vivo, the vector mix containing the 14 barcoded capsids was injected into four highly humanized FRG mice (n = 2 saline control, n = 2 0.6 mg/kg methylprednisolone daily). Methylprednisolone/saline was injected 24 hours prior to AAV injection, and each day after for four days in total. Mice were culled at day 4 post AAV injection and human hepatocytes were sorted as described below. The barcode region was amplified and sequenced as described above. Results are presented in Supplementary Fig. 6.

## Enzyme-linked Immunosorbent Assay (ELISA) measurement of anti-AAV IgG specific antibody titer in human serum

Human sera were assayed for reactivity to all the fourteen capsids by ELISA, following a recently described method[8]. 96-well polystyrene ELISA plates (Nunc #442404) were coated overnight at 4 °C with 50 µL per well of the AAV vector mix ($2.5×10^{10}$ vg/mL) diluted in coating buffer (carbonate-bicarbonate buffer, SigmaAldrich). Plates were washed 3 times with wash buffer (PBS) + 0.05% Tween-20 (Sigma Aldrich) and then received 100 µL per well of blocking buffer (PBS + 5% skim milk + 0.05% Tween-20). Plates were then washed three times after incubation at room temperature for 2 hours in wash buffer and received 50 µL per well of sera (diluted in blocking buffer at 1:50 and at 1:200 with duplicate wells for each dilution). Plates were incubated for 2 hours at room temperature and washed 3 times with wash buffer before receiving 50 µL per well of horse radish peroxidase (HRP)-conjugated anti-human Fc specific IgG (Chemicon AP309P, diluted 1:10,000 in blocking buffer). Plates were incubated for 1 hour at room temperature and washed 4 times using wash buffer before receiving 75 µL per well of 3,3′,5,5′- Tetramethylbenzidine (TMB, Sigma-Aldrich). Plates were incubated in the dark for 30 minutes at room temperature and the reactions were then stopped using 75 µL per well of 1 M sulfuric acid. The absorbance of each well was measured at 450 nm wavelength using a VersaMax microplate reader (Molecular Devices, LLC). Duplicate wells containing no AAV served as background controls. The mean value for each sample dilution was calculated for wells with (foreground) and without coated vector (background) and the sample was considered reactive if this ratio was >2.0. Reactive and non-reactive human sera to the whole mix were subsequently assayed for reactivity to all the fourteen capsids individually following the same protocol. The chosen sera were diluted in blocking buffer at 1:25, 1:50, and at 1:100 with duplicate wells for each dilution.

## Liver origin and procurement

The two livers described in this manuscript were obtained from DonateLife, the centralized donation organization in Australia (Supplementary Table 2). These were unsuitable for transplantation but consented for research use. These livers were accepted in accordance with a study protocol approved by the Sydney Local Health District Ethics Review Committee (X18-0523 & 2019/ETH08964). The first liver was medically unsuitable for transplantation due to biliary sepsis and cholecystostomy, whereas the latter was a donation after circulatory death (DCDD) that did not meet the acceptance criteria for transplantation of age <50. Livers were procured in our standard fashion with aortic flushing using cold Soltran (Baxter Healthcare, Illinois, USA) and University of Wisconsin preservation solution (Belzer UW, Bridge

to Life, Columbia, USA). In addition, DCDD livers received an intra-aortic injection of tissue plasminogen activator (Alteplase 20 mg) delivered as a bolus at the time of cold perfusion. All livers were placed in static cold storage for transfer to our center.

## Liver viability test

At the indicated time points, we assessed both human livers for viability, following the criteria proposed by the VITTAL clinical trial[3]. These are a concentration of lactate ≤2.5 mmol/L, and two or more of: bile production, pH≥7.30, glucose metabolism, hepatic arterial flow ≥150 ml/minute and portal vein flow ≥500 ml/minute, or homogeneous perfusion.

## Liver assist modifications

We utilized the recently described[5] modified commercial liver perfusion system (Liver assist, Organ assist, Gronigen, Netherlands), as we have previously described[13], which uses an open venous reservoir. Briefly, we added a flow adjustable dialysis membrane (Prismaflex or Polyflux, Baxter Healthcare, Illinois, USA) for filtration of water-soluble toxins and control of perfusate volume, long term oxygenators (Quadrox-iD Pediatric, Macquet, Getinge Group, Rastatt, Germany) for extended perfusion, and a gas blender (Device Technologies, Sydney, Australia) with a pediatric flow regulator for fine control of ventilation. The perfusate contained 4 units of human-packed red cells, 2 units of fresh frozen plasma, 200 mL of 20% albumin and 1 L of normal saline. Anticoagulation was maintained using enoxaparin (100 mg twice daily) and nutritional support provided using infusions of amino acids (Synthamin 17, Baxter Healthcare, Illinois, USA), lipids (Clinoleic 20%, Baxter Healthcare, Illinois, USA), taurocholic acid (7.7 mg/h), methylprednisolone (21 mg/h), insulin (Actrapid 2 IU/ml) and glucagon (20 ug/mL).

Prior to connecting the liver to the perfusion machine, we circulated the perfusate and activated the parallel dialysis circuit until the potassium, calcium and acidosis abnormalities in the stored blood were corrected. Specifically, the parallel dialysis circuit used permits filtration of electrolytes across the dialysis filter. This allows the perfusate to be equilibrated with the dialysate to bring potassium, calcium (and other electrolytes) into the intended range.

After flushing with saline, we connected the liver to the system by portal vein and hepatic artery cannulas and we rewarmed the perfusate at 1 degree/h from 32 °C to 36 °C to minimize hemolysis. A flow chart of the modifications can be found on Fig. 1a, and images of the modifications have been published previously by our group[13].

## Liver splitting

The ex situ split was performed the day after the commencement of whole liver perfusion, which corresponded to a whole liver perfusion time of 12-16 hours. Splitting into a left lateral sector graft (LLSG, segments 2 and 3) and an extended right graft (ERG, segments 1 and 4-8) was performed without interrupting arterial or portovenous perfusion and required 2 surgeons and a scrubbed assistant. The procedure itself was divided into 3 phases which have been described in detail recently[24]. Initially, the liver's surface anatomy was clarified by removing surrounding tissues, and the hepatic vessels were prepared for cannulation. A special manoeuvre was used to facilitate the liver's parenchymal transection near the falciform ligament. Further dissection allowed for the isolation of the liver's left and right vascular structures. Indocyanine green (ICG) was used to demarcate the division line, ensuring the LLSG retained maximum functionality and the ERG minimum ischemia. Parenchymal division was carefully carried out using a Harmonic scalpel. Segment 4's vascular structures were isolated and divided. The left hepatic duct was then identified with ICG cholangiography, and the division point was confirmed with choledochoscopy. Finally, the ERG was attached to a secondary perfusion system, with the perfusate prepared to the same specifications as

before. The right hepatic artery and left portal vein were severed and sealed, and reperfusion was initiated on the ERG through a new cannula, while the LLSG continued to receive perfusion through the hepatic artery. Both grafts were then maintained with specific pressure settings to ensure their viability. After the liver split, we ensured that the perfusate fresh frozen plasma (FFP) conditions were maintained consistent with the pre-split conditions.

## Measurements of liver function

Liver synthetic function was assessed by measuring lactate clearance, bile production, and perfusate biochemistry. We measured lactate concentration, pH, glucose concentration, $pO_2$ and $pCO_2$ with a blood gas analyser (RAPIDPoint 500, Siemens Healthengineers, Norwood, Massachusetts, USA). The liver function tests, coagulation studies, and Factor V were measured by the clinical laboratory of the Royal Prince Alfred Hospital, Sydney, Australia. We assessed the architectural integrity of the liver by staining formalin-fixed core biopsies with haematoxylin and eosin. Biopsies were also scored by a specialist pathologist for percentage of coagulative necrosis, hepatocyte detachment, bile duct injury (Supplementary Figs. 8–10).

## Statistics and reproducibility

Nonparametric statistical analyzes were performed using the two-tailed Mann-Whitney test with the specified biological and technical replicates in each experimental group. (*$P ≤ 0.05$; **$P ≤ 0.01$; ****$P ≤ 0.0001$; n.s., $P > 0.05$). For Figs. 3f, g and 4f, g, n = 2 independent biopsies were analyzed, respectively. For Fig. 3h–k and Fig. 4h–k, n = 1 biopsy was analyzed, respectively. The Investigators were not blinded to allocation during experiments and outcome assessment.

## DNA and RNA isolation and cDNA synthesis

To extract DNA, cells or tissue were first suspended in 200 μL of a lysis solution composed of 100 mM Tris-HCl with a pH of 8.5, 5 mM EDTA, 0.2% sodium dodecyl sulfate, and 200 mM NaCl, all of which included 50 μg/mL of proteinase K. This mixture was then incubated at 56 °C throughout the night. After this, PureLink RNase A (Thermo Fisher Scientific, #12091021) was added to the mixture at a concentration of 0.4 μg/μL and the mixture was further incubated at 37 °C. The DNA was subsequently isolated using the classic phenol/chloroform method with a mixture of phenol, chloroform, and isoamyl alcohol (25:24:1, Sigma-Aldrich, #P3803), and then concentrated and purified through ethanol precipitation. For RNA extraction, the Direct-zol kit was utilized (Zymogen, #R2062), followed by treatment with TURBO DNase (Thermo Fisher Scientific, #AM2238) to remove any contaminating DNA. The resulting RNA was used to create cDNA with the help of the SuperScript IV first-strand synthesis system, adhering to the guidelines provided by the manufacturer (Thermo Fisher Scientific, #18091050).

## Immunofluorescence analyzes

Immunofluorescence was performed, as described in detail recently without modifications[6]. Specifically, liver tissue was fixed with paraformaldehyde, cryo-protected in sucrose, and frozen in O.C.T. (Tissue-Tek). Frozen liver sections (5 μm) were permeabilized in ice-cold methanol, then room temperature 0.1% Triton X-100, and then reacted with DAPI at 0.08 ng/mL (Invitrogen, D1306) and the antibodies indicated below. Anti-eGFP (Invitrogen, #A-11122), Anti-HNF4α (Abcam, #ab41898), Anti-HAL (Sigma Prestige Antibodies, #HPA038547), anti-Cytokeratin 7 (Abcam, #ab68459), anti-albumin (Bethyl, #A80229A), anti-KIAA0319L (AAVR, Abcam, #ab105385), anti-CD68 (Invitrogen, #14068882) antibodies were used as indicated. For Figs. 3k, and 4h–k antigen retrieval was required. This was achieved by heating the slides for ten minutes in antigen retrieval buffer (Tris-EDTA, pH 9). After immunolabelling, the images were captured and analyzed on a LSM800-Airyscan microscope using ZEN Black software.

**Mouse studies and isolation of human hepatocytes by collagenase perfusion**

All animal experimental procedures and care were approved by the joint Children's Medical Research Institute (CMRI) and The Children's Hospital at Westmead Animal Care and Ethics Committee. Fah$^{-/-}$Rag2$^{-/-}$Il2rg$^{-/-}$ (FRG) mice were bred, housed (12 hours dark/light cycle; ~50% humidity; 22-23 °C), engrafted, and monitored as recently described[6]. Levels of human and non-human primate cell engraftment were estimated by measuring the presence of human albumin in peripheral blood, using the human albumin ELISA quantitation kit (Bethyl Laboratories, #E80-129). To evaluate the AAV transduction potential, mice were placed on 10% NTBC and were maintained in this condition until harvest. All hFRG mice reported in this study were engrafted with human and non-human primate hepatocytes from the same donors (Caucasian, 15-month-old donor, Lonza, #HUM181791; ThermoFisher, #CY409, respectively).

Mice were randomly assigned to experiments and transduced via intravenous injection (lateral tail vein) with the indicated vector doses in a total volume of 150 μl. Mice were euthanized by $CO_2$ inhalation either 1 week after transduction for barcoded NGS analyses, unless indicated otherwise. To obtain murine and human/non-human primate single-cell suspensions from xenografted murine livers, we followed the same collagenase perfusion procedure as recently described[6]. Briefly, cells were labeled with phycoerythrin (PE)-conjugated anti-human-HLA-ABC (clone W6/32, Invitrogen 12-9983-42; 1:20), biotin-conjugated anti-mouse-H-2Kb (clone AF6-88.5, BD Pharmigen 553,568; 1:100), and allophycocyanin (APC)-conjugated streptavidin (eBioscience 17-4317-82; 1:500). Flow cytometry was performed in the Flow Cytometry Facility, Westmead Institute for Medical Research (WMIR), Westmead, NSW, Australia. The data were analyzed using FlowJo 7.6.1 (FlowJo LLC).

**Non-human primate work**

Animal procedures were approved by the ethical committee for animal testing of the University of Navarra and by the Department of Health of the government of Navarra (Comité de Etica para la Experimentación Animal code: 038/15) and performed according to the guidelines from the institutional ethics commission. Animal welfare checks were performed by animal care staff twice daily. A young adult male *Macaca fascicularis* NHP animal was subjected to the immunoadsorption process (described in detail in[25]). Specifically, before starting immunoadsorption (IA), the non-human primate was administered darbepoetin alpha (100 μg, Aranesp; 0.45 μg/kg, Amgen) alongside ferric carboxymaltose (Ferinject, 10 mg/kg; Vifor Pharma) once a week for a total of three weeks. IA procedures were carried out while the NHP was under anesthesia, which was induced and maintained using a combination of ketamine (Imalgene 1000; Merial), midazolam (B. Braun), propofol (B. Braun), and sevoflurane (AbbVie). The TheraSorb–Ig Flex adsorber filter (Miltenyi Biotec) was employed in conjunction with the IA device Life 18TM (Miltenyi Biotec). A single IA session was conducted, which consisted of three sequential cycles. Within the following 30 min after immunoadsorption, the vector was infused via the saphenous vein over ten minutes. At day 30, the animal was euthanized, and the four liver lobes were collected for further analysis. Neutralization assays determining the anti-capsid antibodies for AAV9 prior to injection were performed as described previously with no major modifications[26]. Specifically, HEK293T cells were plated in a 96-well plate at 10,000 cells per well. On the next day, the NHP serum was diluted in DMEM containing 2% FBS starting at a 1:5 ratio, followed by a series of 1:3 dilutions. Each diluted serum was mixed with AAV9, which was prepped at a concentration of 10,000 viral genomes (vg) per cell and coding for luciferase, and incubated for 2 hours at 37 °C. The mixture was then used to transduce the HEK293T cells, with each serum dilution tested twice. Control groups included non-transduced cells and cells transduced with AAV9 that had not been pre-mixed with NHP serum. After a 48-hour incubation period, luciferase activity in the cells was measured to quantify light emission. The neutralizing antibody (NAb) titer was defined as the highest serum dilution at which light emission was reduced to 50% of the positive control without serum. A serum dilution that decreased AAV9-mediated transduction by 50% or more was marked as positive, and this highest positive dilution indicated the NAb titer.

**Reporting summary**

Further information on research design is available in the Nature Portfolio Reporting Summary linked to this article.

## Data availability

All data generated or analyzed during this study are included in this published article and the Supplementary information files. Raw sequencing data are available via accession code PRJNA1076589. Source data are provided with this paper. All the capsid sequences used in this article have been previously described in the literature (Supplementary Table 1). Any other relevant data are available from the corresponding author upon request. Source data are provided with this paper.

## Code availability

The scripts used for barcode detection are available on Code Ocean (https://doi.org/10.24433/CO.7176285.v1).

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

## Acknowledgements

We thank the Cytometry Facility of the Westmead Institute for Medical Research for help with sorting of human/murine hepatocytes. We also would like to thank all the members of the Children's Medical Research Institute Bioresources facility, with special thanks to S. Dimech. We would also like to thank Dr Razvan Albu for help generating the bar-coded transgenes, Dr Anais Karime Amaya for initial assessment of liver explant viability, and Dr Grant Logan for his guidance on the ELISA procedure described herein. This work was supported by project grants from the Australian National Health and Medical Research Council (NHMRC) to L.L. (APP2021305 and APP1161583). The work of L.L. was also supported by a research grant from the National Science Centre, Republic of Poland (OPUS-21) (2021/41/B/NZ5/01671). M.C-C. was also supported by a 2021 New South Wales (NSW) Ministry of Health, Office of Health and Medical Research (OHMR) Early-Mid Career Research Grant - Gene and Cell Therapy. Financial support was also provided by the Royal Prince Alfred Hospital Transplant Institute. N-G.L is supported by the Australian Government Research Training Program Stipend Scholarship.

## Author contributions

Conceptualization, M.C.-C., I.E.A., and L.L.; Methodology, M.C.-C., N-S.L., M.L., C.U., G.G-A, and C.P.; Investigation: M.C.-C., S.H.Y.L., R.G.N., M.K., D.N., N-S.L., M.L., E.Z., R.B.D., A.F.V., A.W., and G.B.; Writing – original draft, M.C.-C.; writing – review and editing, M.C.-C., I.E.A., and L.L.; Funding acquisition, M.C-C., G.M., C.U., G.G-A, I.E.A., C.P., and L.L.; Visualization, M.C.-C., J.M., R.R.-P.; Supervision, M.C.-C., G.M., C.U., G.G-A, I.E.A., C.P., and L.L.

## Competing interests

L.L. and I.E.A. are co-founders of Exigen Biotherapeutics, a company that utilizes similar technologies broadly discussed in this paper. M.C.-C., I.E.A., and L.L. are inventors on patent applications filed by Children's Medical Research Institute related to AAV capsid sequences and in vivo function of novel AAV variants (WO2021168509). The remaining authors declare no competing interests.

## Additional information

[1]Translational Vectorology Research Unit, Children's Medical Research Institute, Faculty of Medicine and Health, The University of Sydney, Sydney, Westmead, Australia. [2]Australian National Liver Transplantation Unit, Royal Prince Alfred Hospital, Faculty of Medicine and Health, The University of Sydney, Sydney, Australia. [3]Centre for Organ Assessment Repair and Optimisation, Royal Prince Alfred Hospital, Faculty of Medicine and Health, The University of Sydney, Sydney, Australia. [4]Gene Therapy Research Unit, Children's Medical Research Institute and The Children's Hospital at Westmead, Faculty of Medicine and Health, The University of Sydney, and Sydney Children's Hospitals Network, Sydney, Westmead, Australia. [5]Gene Therapy and Regulation of Gene

Expression Department, IdiSNA, Instituto de Investigación Sanitaria de Navarra, Universidad de Navarra, CIMA, Pamplona, Spain. [6]Liver Injury and Cancer Program, Centenary Research Institute, A.W Morrow Gastroenterology and Liver Centre, Sydney, Australia. [7]Discipline of Child and Adolescent Health, The University of Sydney, Sydney Medical School, Faculty of Medicine and Health, Sydney, Westmead, Australia. [8]Australian Genome Therapeutics Centre, Children's Medical Research Institute and Sydney Children's Hospitals Network, Sydney, Westmead, Australia. [9]Military Institute of Medicine - National Research Institute, Laboratory of Molecular Oncology and Innovative Therapies, Warsaw, Poland. ✉e-mail: llisowski@cmri.org.au

