## [Peer Review File · Nature Communications]

Reviewers' Comments:

Reviewer #1:

Remarks to the Author:

The study by Marti et al. introduces a pioneering approach to expand the efficacy models of AAV transduction. The researchers successfully established an ex situ liver perfusion platform using human liver, allowing the screening of AAV transduction and blocking neutralizing antibodies (Nab) in a physiologically relevant setting with native liver anatomy. The data collected are informative and compelling, providing valuable insights that could serve as a new reference for the clinical application of AAV gene therapy. However, the reviewers have raised below points that require addressing.

Major Revisions:

1. The primary focus revolves around assessing the dependability and viability of utilizing "whole human liver explants in an ex situ normothermic perfusion system" for AAV pre-clinical screening. This arises due to challenges such as the scarcity of available whole human livers, variations in genetic background, and the potential influence of perfusion-related cell stress on AAV transduction. It's evident that the existing procedure falls short in evaluating long-term results. Furthermore, the dissimilarities in data between liver explants and xenografted models undermine the persuasiveness of this model.
2. The removal of multiple biopsies for pathological checks and transgene surveys might impact the stability of the established perfusion flow, potentially affecting the biodistribution of AAV vectors. The authors should address this concern and provide evidence supporting the stability of the perfusion flow during the removal procedure.
3. Considering the potential impact of steroid enhancement on AAV transduction, the authors should discuss the potential selective pressure on AAV tropism and transgene expression when using the AAV mixture with 14 serotypes.
4. The claim that "specific capsids effectively obstructed liver uptake of those" AAV-LK03 and AAV-SYD12, attributing it solely to the presence of Nab without cross-reactivity, might not be fair in the context of different genetic backgrounds from Donor1 and 2. The authors should provide a more comprehensive discussion on this issue.
5. To enhance the manuscript's value for readers, the authors should consider providing live videos or images showcasing the modified perfusion platform (Fig 1A).
6. Fig 1B: Either the sequence or the secondary structure from 44-mer barcode may influence the AAV package and transduction, although this work selected the equivalent BC by screening on AAV2 and D/J, the potential bias on various capsids should be also considered. And the interaction impact could make the distribution and tropism different from the original vector w/o BC, which further tends to drawing mistake information for the 14 serotypes. Also, reviewer is curious that the underlying cause on molecular equity in mixture of AAV when they are confirmed by NSG reads of BC, did each serotype are packaged individually?
7. It would be beneficial to include correlation indices between transgene expression and host vg, presented as the "quotient of NGS reads from cDNA and viral gDNA from host cells" in figures or tables, to better understand the efficacy of AAV delivery.
8. The unmatched data between perfused human livers and humanized liver challenge the reliability of in situ models compared to in vivo models. The authors should address this issue and consider testing AAV transduction using mimic mouse liver perfusion models to validate in vivo data.
9. The authors should clarify whether the data in Fig 4 d & e are from sorted human cells or whole liver cells without any selection.
10. Considering to use the same donors for the in situ models as for the engrafted mice and monkeys models could provide reliable and comparable evidence. This could be possible since the liver splitting procedure has already been accomplished.
11. What is the replacement indices in engrafted monkey liver?

Minor Revisions:

12. In sFig1, the method description should be provided, and the expression index needs clarification. Additionally, the number of replicates (N=3) and their details should be specified.

Expression index: is it from reads of BC/ vg from host cells or vg from "the vector preparation"?
N= 3 is for each transduction from MOI=50, 500 and 5000 vg/cell, or triplicates for certain MOI?
13. On page 26, the reference for "as previously described" should be included.
14. In sFig 4-8, "ER" should be corrected to "ERG." to match the figure legend.
15. The color scheme for whole and LLSG in ALT measurement is too similar, making it difficult to identify them. As ALT is an essential marker for monitoring clinical adverse reactions on AAV hepatotoxicity, the authors should discuss the implications of the difference in ALT response in ERG between Donor1 and 2 and provide suggestions to avoid such changes.
16. Additional information, such as the weight of the human liver, BMI, etc., should be provided if possible.

Reviewer #2:

Remarks to the Author:

Cabanes-Creus and colleagues present data from an ambitious study using normothermic human ex-situ liver explants as a new method to evaluate/screen for the short-term efficacy of AAV vectors. Experiments were carried out in two liver explants treated with an AAV-GFP constructs in 14 different capsids via direct portal vein infusion, in conditions aiming to mimic AAV antibody negativity or positivity. This model provides closer representation of liver biology to other in vitro systems for evaluation of gene therapy vectors, albeit lacking circulating immune cells and other immune tissues. The manuscript presents parallel data from other pre-clinical models consisting of non-human primate and FRG xenograft (NHP & human hepatocyte engrafted) murine models. The paper is novel and demonstrates an interesting approach that could have significant implications for evaluating short term (< 1 week) uptake and efficacy of viral and non-viral vectors. With the volume of data presented the overarching aim and messages are sometimes difficult to follow. Although well written, the manuscript is overly long and would benefit from significant editing to help provide a clearer message. I have the following main comments:

Major

1. Study Ethics: Within the methods, there is a statement that the organ donors provided consent for organ usage for research purposes. Given the complexity and potential ethical questions this model may raise, I would like to see a statement of ethical reviews and approvals for this study protocol in the main manuscript. It may be interesting to provide a brief comment on the ethical questions raised by this model compared to other models in the discussion?

2. Results: Non-Neutralising Ex-Situ Liver: For the first liver explant, there is an early acute elevation (day 2) post-split in IL6 and ALT for the ERG compared to the LLSG. This effect seems more than was seen for donor 2? Could this indicate a more significant reperfusion injury in this graft? Please provide comment on the degree of cell death on the biopsies (supplementary figure 20 – 22) and whether this may have contributed to any relative decline in transduction seen for some of the AAV capsids between the different grafts. Was there any sign of cellular stress post transduction?

3. Results: Non-Neutralising Ex-Situ Liver: The data in these experiments shows a relative increase in VCN for the LLSG compared to the ERG. The transduction data in figure 2d appears quite heterogenous between the capsids with some decreases for the LLSG and some increases seen for the ERG. Please provide a clearer summary to which capsids contribute to this elevation for the LLSG. Were the liver biopsies single or multiple cores?

4. Results Non-Neutralising Ex-Situ: Please provide comment on the hepatic location of GFP expression within the liver. In panels 2f & 2g, expression appears visually more in the ERG than the LLSG?

5. Results & Methods: AAV antibody assays: Assays of AAV antibodies are far from standardised and information presented is based on two previous publications. The paper cited from Gardner et al studied correlation for AAV1, 8 and 9 antibodies only. These findings are extrapolated to the multiple different AAV capsid antibodies evaluated in the ELISA developed for this study. Given one of the key questions is whether an effect is seen from inhibitory AAV antibodies on transduction in the ex vivo model, I would have preferred to see a functional transduction inhibition assay,

although recognise that this may no longer be possible. Within Figure 3 (panel A), please provide brief overview of AAV antibody reactivity (e.g., negative, 1:25 & 1:50).

6. Results: Neutralising Plasma Ex-Situ Liver: The authors hypothesise that there was uptake by Kupffer cells of opsonised antibodies, although it was not possible to demonstrate this using IHC. Within this model could these complexes have been cleared on the dialysate membrane?

7. Results: Neutralising Plasma Ex-Situ Liver: The staining pattern for GFP in the Left lobe in neutralising conditions appear more marked than the first donor, is this a factor of difference in underlying donor characteristics or organ viability?

8. Results: NHP & Murine Models: I wonder if it would be helpful to have a supplementary table or figure summarising the similarities and differences in different models as this section is quite difficult to follow ? In the NHP model, AAV9 transduction is reasonable despite the AAV at a titre of 1:30 compared to donor 1 – Is this a limitation of the NHP model?

9. Discussion: With the different AAV capsids used in the manuscript it would be helpful to have some discussion of the differences/similarities between these capsids, either within the discussion or possibly as an extension of supplementary table 1.

10. Discussion: Vector Mixture Competition: Within the mixture do the authors think there could be competition of vectors for uptake or a potential for antigenic competition with circulating AAV antibodies?

Minor

1. Figure 1: Typo : “Dyalysis” – text box below the hepatic artery pump

2. Figure 2b & 3b : The different AAR serotypes are quite difficult to distinguish – Consider reducing thickness of the data point outlines or adjust colours?

3. Discussion: Limitation : The system lacks circulating immune cells & other potential mechanisms of antibody mediated clearance

4. Methods: Murine xenografts: please provide details of volumes and flow rates – did these injections represent hydrodynamic delivery ?

5. Methods: Perfusate Media & Vectors: The perfusate media contains corticosteroids. There is some discussion as to whether treatment with corticosteroids could facilitate transduction, e.g., BMN 270-303 study in hemophilia A. Could this impact on uptake. Was any estimate made of full: empty capsids for the different constructs?

Reviewer #3:

Remarks to the Author:

Without a doubt, the concept of the study presented by Cabanes-Creus et al is milestone both for NMP-based research as well as AAV development. It exemplifies the potential of this translational approach as it is closing a gap between stem cell and organoid based research, animal models and clinical trials.

The title is well chosen in the sense that the study indicates the potential of the methods applied in this study. However, while the study does successfully describe the feasibility of the concept it comes short with respect to the actual realization of the idea.

With respect to the sequence of experiments, the authors work backwards when they go from the work in human livers to the mouse and cell culture work. It might be advantageous to structure this in reverse order since the AAV based transduction in human livers would be the last rather than the first step.

The group has previously published their model for extended human liver preservation. In their paper (Nature Comm 2023:14:4755) they have described the model of prolonged full and split liver preservation with mixed results.

While the concept of this study is potentially groundbreaking, several aspects of the study represent significant limitations in the overall impact of the work.

The key limitation is the limited number of organs included in the NMP work. Only two livers were used for these experiments. Considering the major inter-organ and perfusion variations, the findings need to be seen in the light of this limitation. One liver was excluded from transplantation due to biliary sepsis and the second liver stemmed from a DCD. Hence the impact of both the infection and the warm ischemia may be relevant and interfere with the findings.

The continuous rise of AST in the perfusate indicates a progressive injury in addition to the I/R injury. This may be a limiting factor for the interpretation of the results. The split liver model, while attractive in terms of a case matched approach, has some significant limitations possible relevant in the context of the study. The two very different graft sizes and the diverse perfusion protocols and flows in the respective livers may influence the course of the experiment. The determination of the viability and function of livers as described in their previous study and as applied here has some limitations and defines a relatively liberal view on what can be considered viable and functional. The histological findings in the previous publication indicate coagulative necrosis between 0 and 100%. The different degrees of necrosis are also present in this study and seem to vary depending on the sampling site (supplementary fig 20). This inconsistency is also illustrated by the different AST and IL-6 levels in the perfusate.

In summary this all speaks towards the limitation arising from the small number of livers and the variables added through splitting.

The second liver received double the dose of AAV compared to the first liver. This limits the ability to compare the two experiments. A weight adjusted dosing seems more reasonable than such an approximation in the dosing.

The mechanisms involved in the anti-AAV neutralization remains speculative. The rapid vector removal is pointing in this direction, but further evidence for the relation is needed.

I struggle to follow and understand the AAV transduction and transgene expression/function in the various models. It is difficult to understand the patterns and the mechanisms behind this. At current, this is rather confusing and not culminating to a coherent picture and understanding.

As the authors point out in the opening reference in their discussion, the study more illustrates the potential of the methods applied rather than a fully conclusive assessment and implementation of these technologies. Several limitations are obvious and should be addressed now and in future studies. Nevertheless, the value and the visionary spirit of this research group deserves recognition.

Minor:

Why were some barcoded constructs packaged in AAV2 and AAV-DJ underexpressed compared to mean transduction values? Why does this indicate successful transduction? (Suppl Fig 1).

The element of tropism in human livers is not addressed sufficiently. A more detailed analysis e.g. using confocal microscopy or single cell RNA analysis would help to achieve this. The mouse data are not conclusive and not immediately transferable to human livers. Again, an n=2 this might not be sufficient to conclusively demonstrate this.

A figure showing histology of all livers at all time points would be helpful to compare the findings between livers/groups.

REVIEWER COMMENTS

Before addressing the received comments, we wish to extend our sincere gratitude to the reviewers for their invaluable time, dedication, and thoughtful feedback on our manuscript. The authors are committed to thoroughly addressing all raised concerns to the best of our ability.

Reviewer #1 (Remarks to the Author):

The study by Marti et al. introduces a pioneering approach to expand the efficacy models of AAV transduction. The researchers successfully established an *ex situ* liver perfusion platform using human liver, allowing the screening of AAV transduction and blocking neutralizing antibodies (Nab) in a physiologically relevant setting with native liver anatomy. The data collected are informative and compelling, providing valuable insights that could serve as a new reference for the clinical application of AAV gene therapy.

We are glad that our data are seen as both informative and compelling, and we hope that our efforts will indeed serve as a valuable reference for the advancement of clinical applications of liver-targeted gene therapies. We greatly appreciate the Reviewer's comments and suggestions and have made several key modifications to address the concerns raised.

However, the reviewers have raised below points that require addressing.

Major Revisions:

1. The primary focus revolves around assessing the dependability and viability of utilizing "whole human liver explants in an *ex situ* normothermic perfusion system" for AAV pre-clinical screening. This arises due to challenges such as the scarcity of available whole human livers, variations in genetic background, and the potential influence of perfusion-related cell stress on AAV transduction. It's evident that the existing procedure falls short in evaluating long-term results. Furthermore, the dissimilarities in data between liver explants and xenografted models undermine the persuasiveness of this model.

We acknowledge the concerns raised by the Reviewer and have incorporated these into the revised discussion. We appreciate the emphasis on the dependability and viability of using whole human liver explants in an *ex situ* normothermic perfusion system for AAV pre-clinical screening. In the revised manuscript, we have included the Reviewer's thoughts on the scarcity of human liver explants and genetic variability. We also discuss the potential influence of perfusion-related cell stress on AAV transduction. While acknowledging these concerns, we still believe that this study serves as an initial proof of concept that advances the field. We recognize that further work with additional livers will be required to fully establish this model's predictive value. From our point of view, the liver explant is an important addition to the preclinical tool-shed for AAV-based technologies, but also for other advanced therapeutics. Due to cost, low accessibility and ethical concerns, it will

probably be even less commonly used than non-human primates (NHPs). Also, similarly to NHPs it is affected by large heterogeneity within the population, which on one side is a limitation, but on the other hand – it reflects the heterogeneity within the patient population and thus arguably better recapitulates the actual clinical reality.

Moreover, we address the disparities between liver explant data and xenograft models as described below, proposing that these differences do not necessarily undermine the model's persuasiveness. Instead, they may point to limitations within the xenograft models themselves. As such, we believe this work lays the groundwork for future studies that will contribute to a more thorough understanding and refinement of preclinical models for human liver studies.

2. The removal of multiple biopsies for pathological checks and transgene surveys might impact the stability of the established perfusion flow, potentially affecting the biodistribution of AAV vectors. The authors should address this concern and provide evidence supporting the stability of the perfusion flow during the removal procedure.

We thank the Reviewer for highlighting the potential impact of biopsy procedures on the stability of perfusion flow and the biodistribution of AAV vectors. In our revised manuscript, we address this concern by specifying that core biopsies had no noticeable effect on perfusate flow (**Page 7**). We monitored the perfusion flow, ensuring that it remained constant despite the removal of multiple biopsies. Core biopsies were systematically taken, and liver wedges were only extracted at specific, predetermined time points to minimize any potential disturbance to the perfusion flow. We believe that this approach allowed us to maintain a stable environment for the assessment of the AAV vectors' biodistribution in the human livers.

3. Considering the potential impact of steroid enhancement on AAV transduction, the authors should discuss the potential selective pressure on AAV tropism and transgene expression when using the AAV mixture with 14 serotypes.

We have now included a supplementary data exploring the possible selective pressures exerted by steroid use on AAV tropism and transgene expression (**Supplementary Figs 5-6**). Briefly, we have studied how the administration of steroids might favour certain serotypes over others, potentially skewing the transduction profile of individual AAVs included in our mix of 14 serotypes. Specifically, we have analysed the tropism and transgene expression profiles of all serotypes both in the presence and absence of steroids, both *in vitro* (**Supplementary Fig. 5**) and *in vivo* (**Supplementary Fig. 6**). No major differences in transduction efficiency were noted between the methylprednisolone and control groups. These findings are briefly discussed on **Page 6** of the manuscript.

4. The claim that "specific capsids effectively obstructed liver uptake of those" AAV-LK03 and AAV-SYD12, attributing it solely to the presence of Nab without cross-reactivity, might

not be fair in the context of different genetic backgrounds from Donor1 and 2. The authors should provide a more comprehensive discussion on this issue.

We have modified the discussion to take into account the valuable points the Reviewer raised. The modified section reads as follows:

'While we observed a notable correlation between the presence of neutralizing antibodies and the obstruction of liver uptake of specific capsids, we must acknowledge the potential impact of specific donor-related factors on our findings. Specifically, one of the livers was excluded from transplantation due to biliary sepsis, and the second liver was procured from a donor after circulatory death (DCD). Hence, the impact of both the infection and the warm ischemia may be relevant and could affect vector performance, adding a level of complexity to the interpretation of the results of our study. Furthermore, with only two livers utilized, our sample size is too small to account for the heterogeneity inherent in human liver physiology. Factors such as varying genetic backgrounds between both donors, as well as age, sex, and disease states, could potentially influence individual capsid performance. These variables underscore the significance of the limitation given the complex nature of liver functions and the variable response to normothermic machine perfusion (NMP) based on individual liver pathology and history. These constraints are critically highlighted in the context of our results and should be carefully considered.¹³ To achieve broader applicability and address these limitations, further research with a larger and more diverse range of liver organs is necessary to validate and generalize our findings.'

5. To enhance the manuscript's value for readers, the authors should consider providing live videos or images showcasing the modified perfusion platform (Fig 1A).

We agree with this comment, however, we have previously published the modifications and provided images (Fig.6 in <https://www.nature.com/articles/s41467-023-40154-8>). We have now added this citation in the corresponding Materials and Methods section.

6. Fig 1B: Either the sequence or the secondary structure from 44-mer barcode may influence the AAV package and transduction, although this work selected the equivalent BC by screening on AAV2 and D/J, the potential bias on various capsids should be also considered. And the interaction impact could make the distribution and tropism different from the original vector w/o BC, which further tends to drawing mistake information for the 14 serotypes.

We appreciate the Reviewer's insight regarding the potential impact of the 44-mer barcode on AAV packaging and transduction. In response to this concern, we have now included additional testing with the AAV-SYD12 capsid, both *in vitro* and *in vivo* (**Supplementary Fig. 3**). These tests have yielded similar expression results when compared with AAV2 and AAV-DJ (**Supplementary Fig. 2**), reinforcing the reliability of our data across different capsids. We acknowledge that variability in capsid structure could influence expression; however, the breadth of our testing is constrained by practical limits. Despite this, our results indicate that the presence of our specific barcodes does not

introduce a noticeable expression bias for the three capsids tested, which is a good indication that the same would be true for other capsids.

It is our belief that the observed biodistribution is primarily influenced by the capsid rather than the transgene itself. We are not employing the transgene without a barcode in our current study. We believe that **Supplementary Fig. 2 and 3** provide enough evidence that any observed differences in transduction and biodistribution can be attributed to the capsid properties rather than variations in the transgene sequence.

Also, reviewer is curious that the underlying cause on molecular equity in mixture of AAV when they are confirmed by NSG reads of BC, did each serotype are packaged individually?

Yes. The procedure is described in Materials and Methods (**Page 29**):

'Briefly, each capsid serotype was co-transfected individually with the corresponding unique barcoded transgene. The resulting cell lysates were tittered individually and subsequently mixed at equimolar ratio. The vector mix was then purified using double Cesium Chloride (CsCl) gradient ultracentrifugation following the previously published protocol.'

We have also incorporated this into the main text for clarification (**Page 6**):

'Each vector preparation was tittered individually, combined at a 1:1 molar ratio, and the composition of the transgene mix validated using NGS.'

7. It would be beneficial to include correlation indices between transgene expression and host vg, presented as the "quotient of NGS reads from cDNA and viral gDNA from host cells" in figures or tables, to better understand the efficacy of AAV delivery.

We have incorporated this request in **Supplementary Fig. 21**. Results are also now mentioned in the main text (**Page 13**).

8. The unmatched data between perfused human livers and humanized liver challenge the reliability of in situ models compared to in vivo models. The authors should address this issue and consider testing AAV transduction using mimic mouse liver perfusion models to validate in vivo data.

We acknowledge the Reviewer's concern about the unmatched data between perfused human livers and humanized liver models and its implications for the reliability of *in situ* models in comparison to *in vivo* models. While testing AAV transduction using mimic mouse liver perfusion models to validate *in vivo* data could indeed be informative, we must convey that such an experiment extends beyond the scope and resources of the current study. The undertaking of mouse liver perfusion models requires additional time and resources that are not available within the constraints of this review's timeframe. However, we recognize the value of such studies and suggest that they would be a worthwhile direction for future research to corroborate the findings presented here.

9. The authors should clarify whether the data in Fig 4 d & e are from sorted human cells or whole liver cells without any selection.

This section has now been moved to Supplementary Results. We have specified, in **Supplementary Fig. 28**, that the presented results were from sorted human hepatocytes.

10. Considering to use the same donors for the *in situ* models as for the engrafted mice and monkeys models could provide reliable and comparable evidence. This could be possible since the liver splitting procedure has already been accomplished.

We appreciate the Reviewer's suggestion to use the same donors for both *in situ* models and engrafted mouse models. Utilizing liver splits from the same donor for different models is indeed a compelling idea that could enhance the comparability of data across systems. It will indeed be feasible to conduct this study, where one lobe from the split liver could be used for transduction studies while the other could be used for perfusion and subsequent hepatocyte recovery, with the aim of engrafting these cells into mice. This approach would provide a more integrated understanding of AAV vector transduction across different biological systems. However, as mentioned before, the execution of such study is beyond the scope of the current review period. Nonetheless, we will consider this methodology for future research as it aligns well with our ongoing efforts to refine and validate *ex situ* liver models.

11. What is the replacement indices in engrafted monkey liver?

This information is provided in **Supplementary Fig. 27**, which reads: 'We injected highly engrafted animals (n=2) with albumin levels 10 mg mL⁻¹ blood (equivalent to estimated >80% repopulation level)'.

Minor Revisions:

12. In sFig1, the method description should be provided, and the expression index needs clarification. Additionally, the number of replicates (N=3) and their details should be specified. Expression index: is it from reads of BC/ vg from host cells or vg from "the vector preparation"? N= 3 is for each transduction from MOI=50, 500 and 5000 vg/cell, or triplicates for certain MOI?

The expression index refers to the quotient between NGS reads mapped at the cDNA level and NGS reads mapped at the DNA level, for each independent barcode.

A clarification has been added in what is now **Supplementary Fig. 2**. 'The results presented herein show the quotient of NGS reads from mRNA and gDNA. N=3 multiplicities of transduction (n=1 per MOT).'

13. On page 26, the reference for "as previously described" should be included.

This reference has now been included (**Ref. 21**, PMID 26222983).

14. In sFig 4-8, "ER" should be corrected to "ERG." to match the figure legend.

We thank the Reviewer for identifying this error. The necessary correction has been made.

15. The color scheme for whole and LLSG in ALT measurement is too similar, making it difficult to identify them. As ALT is an essential marker for monitoring clinical adverse reactions on AAV hepatotoxicity, the authors should discuss the implications of the difference in ALT response in ERG between Donor1 and 2 and provide suggestions to avoid such changes.

We have updated the graphical representation for ALT levels and have applied the same changes to all other parameters under study for consistency and clarity. We have elaborated on the potential causes of elevated ALT within our model in the discussion section as stated below, positing that reperfusion injury is the most plausible explanation for the observed elevation. This is in contrast to clinical trial observations where ALT/AST elevation is often a result of cellular immune responses—a factor not applicable to this human liver explant system. The relevant section on **Page 22** now reads:

*'Nonetheless, a posteriori analysis of the samples revealed a substantial release of alanine aminotransferase (ALT) (**Supplementary Figs 12 and 16**). Considering the fact that these increases were already evident prior to vector injection and in samples collected from the control untreated right graft of donor 2, which was not exposed to the AAV vector (**Fig. 4a**), it is reasonable to attribute the elevated ALT to reperfusion injury. Reperfusion injury is a significant challenge in liver transplantation¹¹ and, in this instance, was likely unrelated to the vector transduction process. In our study, we incorporated the use of methylprednisolone to limit reperfusion injury and associated adverse effects. Nonetheless, future research is necessary to pinpoint the most effective treatment regimen.'*

16. Additional information, such as the weight of the human liver, BMI, etc., should be provided if possible.

We have added the requested information in **Supplementary Table 2**.

Reviewer #2 (Remarks to the Author):

Cabanes-Creus and colleagues present data from an ambitious study using normothermic human ex-situ liver explants as a new method to evaluate/screen for the short-term efficacy of AAV vectors. Experiments were carried out in two liver explants treated with an AAV-GFP constructs in 14 different capsids via direct portal vein infusion, in conditions aiming to mimic AAV antibody negativity or positivity. This model provides closer representation of liver biology to other in vitro systems for evaluation of gene therapy vectors, albeit lacking circulating immune cells and other immune tissues. The manuscript presents parallel data from other pre-clinical models consisting of non-human primate and FRG xenograft (NHP & human hepatocyte engrafted) murine models. The paper is novel and demonstrates an interesting approach that could have significant implications for evaluating short term (< 1 week) uptake and efficacy of viral and non-viral vectors. With the volume of data presented the overarching aim and messages are sometimes difficult to follow. Although well written, the manuscript is overly long and would benefit from significant editing to help provide a clearer message. I have the following main comments:

We thank the Reviewer for recognizing the ambition and novelty of our study. The Reviewer's positive remarks in regards to the potential implications of this model for evaluating the short-term efficacy of viral and non-viral vectors are greatly appreciated. We acknowledge the feedback regarding the length of the manuscript. In response to the comments regarding the organization and focus of the manuscript, we have now moved the results and discussion pertaining to the alternative liver models to the supplementary information section. This change allows us to concentrate the main body of the paper on the novel aspects of the human liver explant system, thus clarifying the central messages and significantly streamlining the narrative for the reader.

Major

1. Study Ethics: Within the methods, there is a statement that the organ donors provided consent for organ usage for research purposes. Given the complexity and potential ethical questions this model may raise, I would like to see a statement of ethical reviews and approvals for this study protocol in the main manuscript. It may be interesting to provide a brief comment on the ethical questions raised by this model compared to other models in the discussion?

We apologize for not including previously the study ethics on the Materials and Methods section. It now reads (**Page 33**):

'These livers were accepted in accordance with a study protocol approved by the Sydney Local Health District Ethics Review Committee (X18-0523 & 2019/ETH08964).'

As per the Reviewer's comment, we have also added a brief discussion on the ethical aspects of this model in the Discussion section, which now reads (**Pages 21/22**):

'Given the complexity and potential ethical questions surrounding this model, it is imperative to emphasize that the primary intent for donated livers is transplantation, aiming to provide

direct lifesaving benefits to recipients in need. However, in instances like those described here, where livers were unsuitable for transplantation, redirecting these organs for research use offers an alternative ethical use of those organs, contingent upon the presence of an approved human research protocol and an appropriate informed consent.'

2. Results: Non-Neutralising Ex-Situ Liver:

For the first liver explant, there is an early acute elevation (day 2) post-split in IL6 and ALT for the ERG compared to the LLSG.

This effect seems more than was seen for donor 2? Could this indicate a more significant reperfusion injury in this graft?

We acknowledge the Reviewer's observation regarding the acute elevation of IL6 and ALT in the first liver explant by day 2 post-split, which indeed suggests a more pronounced reperfusion injury when compared to the second donor's explant. We concur that the second liver exhibited greater stability and viability, as evidenced by its longer functional duration within the perfusion system. In line with this observation, we have included additional data in the manuscript that highlights the superior functionality of the second liver explant. This is exemplified by the significant increase in Factor V synthesis, which serves as a marker of liver function, from 14% at the onset of perfusion to over 100% for both the Left Lobe Split Graft (LLSG) and the Entire Right Graft (ERG) by five days post-split, as detailed in **Supplementary Figure 15**. This enhancement in liver functionality aligns with the more stable biomarker readings and provides further evidence for the second graft's robustness post-reperfusion.

Please provide comment on the degree of cell death on the biopsies (supplementary figure 20 – 22) and whether this may have contributed to any relative decline in transduction seen for some of the AAV capsids between the different grafts. Was there any sign of cellular stress post transduction?

The degree of observed cell death was moderated and relatively stable over time (**Supplementary Figs 7-10**). We have now provided also a representative histology image for each time point, for each of the lobes (**Fig. 2d, Fig. 2h, Supplementary Figs 7, 14**).

Our analysis suggests that the relative decline in transduction between graphs correlates more closely with the presence of neutralizing antibodies rather than cell death. If cell death were the primary cause, it would likely affect all capsids non-selectively, which was not the case. We did not perform direct studies on cellular stress post-transduction, however, the second liver's right graft, which did not receive AAV treatment, demonstrated a longevity and stability in perfusion comparable to the treated grafts. This observation further supports the hypothesis that the elevation in biomarkers is attributable to reperfusion injury rather than transduction-related cellular stress. Moreover, the dose of AAV used in our experiments was relatively low, which reduces the likelihood of dose-dependent cellular toxicity contributing significantly to the observed results.

3. Results: Non-Neutralising Ex-Situ Liver: The data in these experiments shows a relative increase in VCN for the LLSG compared to the ERG. The transduction data in figure 2d appears quite heterogenous between the capsids with some decreases for the LLSG and some increases seen for the ERG. Please provide a clearer summary to which capsids contribute to this elevation for the LLSG.

We have now provided a clearer summary of which capsids contribute to this elevation. The relevant text **on Page 11** reads:

'We found an increase in the average vector copy number in the left graft (LLSG) that continued to be perfused with the AAV-containing perfusate following the organ split (Fig. 3c). In contrast, the vector copy number in the right graft, which received AAV-free perfusate after the organ split, remained relatively stable at the tested timepoints, ranging from 1.5 to 1 vg per haploid genome (Fig. 3c).'

'The transduction data we obtained from the left graft (LLSG) appeared more variable than the data for the right graft (refer to top panel of Fig. 3d). We believe this was likely a consequence of vector recirculation in the perfusate following organ split, which facilitated ongoing transduction during the course of the study. Specifically, in the LLSG, we detected increased relative transduction of some variants over time (AAV-SYD12, AAV-LK03, and AAV-SEQ3), while others exhibited decreased relative transduction over time (AAV-FT01, AAV-hu.Lvr06, AAV-LK03-REDH). Transduction for the remaining vectors remained stable. Since these data present relative transduction, a decreasing contribution in the liver biopsies likely indicates either slower transduction kinetics or a decreasing amount of artificial signal stemming from the interstitial perfusate, rather than from vector uptake in hepatocytes. Indeed, the vectors showing a decreased relative transduction over time are the same one that stayed longer in the perfusate (Fig. 3c).'

Were the liver biopsies single or multiple cores?

We took multiple biopsies at each time point, for histology, vector copy number, and NGS analyses. For NGS analyses, the results at each time point correspond to a single biopsy.

4. Results Non-Neutralising Ex-Situ: Please provide comment on the hepatic location of GFP expression within the liver. In panels 2f & 2g, expression appears visually more in the ERG than the LLSG?

We added the comment below to the main manuscript (**Page 13**):

'Notably, there was an absence of distinctive vector zonation, which could be attributed to the diversity of capsids present in the vector preparation utilized.'

It is important to add that the observations reported are based on a single wedge biopsy from each graft at the determined time point. We wanted to minimize potential injury from sampling. The apparent greater expression in the Entire Right Graft (ERG) compared

to the Left Lobe Split Graft (LLSG) observed in panels (now **3f & 3g**) could be stochastic, reflecting partial variability of transduction efficiency across different regions of the liver. Given the low levels of transduction and the use of a mix of capsids, it is challenging to determine which specific capsids are contributing to the GFP expression at the protein level from these biopsies alone.

5. Results & Methods: AAV antibody assays: Assays of AAV antibodies are far from standardised and information presented is based on two previous publications. The paper cited from Gardner et al studied correlation for AAV1, 8 and 9 antibodies only. These findings are extrapolated to the multiple different AAV capsid antibodies evaluated in the ELISA developed for this study. Given one of the key questions is whether an effect is seen from inhibitory AAV antibodies on transduction in the ex vivo model, I would have preferred to see a functional transduction inhibition assay, although recognise that this may no longer be possible. Within Figure 3 (panel A), please provide brief overview of AAV antibody reactivity (e.g., negative, 1:25 & 1:50).

We recognize the inherent variability in AAV antibody assays and understand the Reviewer's concerns regarding the standardization of these assays. Accordingly, we have revised our manuscript to remove the reference to the correlation from Gardner et al. due to its limited scope on only a few AAV capsids. We agree with the Reviewer that a functional transduction inhibition assay would have been informative. However, as pointed out, the performance of these capsids *in vitro* can be markedly different, which presents its own set of limitations. While such an assay is not feasible at this stage of our research, we have confidence in the robustness of the ELISA developed for our study, which we believe allows for the assessment of reactivity without bias. As requested, we have now included a brief overview of AAV antibody reactivity in what has been updated to **Figure 4a**. This overview categorizes the reactivity levels (negative, 1:25, and 1:50) to provide clearer insights into the antibody profiles encountered in our study.

6. Results: Neutralising Plasma Ex-Situ Liver: The authors hypothesise that there was uptake by Kupffer cells of opsonised antibodies, although it was not possible to demonstrate this using IHC. Within this model could these complexes have been cleared on the dialysate membrane?

The cut-off of the membrane is 60 kDa, which is substantially below the approximately 4751 kDa size of full AAV particles as estimated previously (Zoratto et al, 2021: PMID: 34608711). Therefore, we think that is unlikely that the complexes were cleared on the dialysate membrane. We have modified the main text accordingly (page X):

'Taking into account the fact that the vector particles reactive with anti-AAV neutralizing antibodies (Nabs) present in the plasma were eliminated from this closed circulation (since they were undetectable in the DNA purified from biopsy samples as well as in the perfusate) and considering that the dialysis filters used had a cut-off of 60 kDa, which is substantially below 4751 kDa estimated size of full AAV8 particles⁹, thereby precluding their filtration, we

explored the possibility that Kupffer cells (the resident macrophages in the liver) were responsible for this active vector clearance.'

7. Results: Neutralising Plasma Ex-Situ Liver: The staining pattern for GFP in the Left lobe in neutralising conditions appear more marked than the first donor, is this a factor of difference in underlying donor characteristics or organ viability?

The more pronounced GFP staining pattern observed in the left lobe of the second donor under neutralizing conditions could indeed be reflective of variability in organ characteristics or differences in donor-derived organ viability. With a sample size of two (n=2), it is challenging to derive definitive conclusions about the influence of underlying donor characteristics on transduction efficiency. We have acknowledged this variability as a limitation within the discussion section of our manuscript. This acknowledgment underscores the need for larger sample sizes in future studies to better understand donor variability and its impact on gene transduction in human liver explants.

8. Results: NHP & Murine Models: I wonder if it would be helpful to have a supplementary table or figure summarising the similarities and differences in different models as this section is quite difficult to follow?

We thank the Reviewer for this suggestion and we have now added a table summarizing the main characteristics of the models and the transduction results (**Supplementary Table 4**). As mentioned before, we have relocated the results and discussion pertaining to the alternative liver models to the supplementary information section.

In the NHP model, AAV9 transduction is reasonable despite the AAV at a titre of 1:30 compared to donor 1 – Is this a limitation of the NHP model?

We acknowledge the Reviewer's note on the differences in AAV9 transduction between the NHP model and donor 1. While there is a differential transduction profile, we believe this does not necessarily reflect a limitation of the NHP model. Direct comparisons between these models are challenged by several limitations, particularly the different assays used to assess antibody titers. For the human sera, an ELISA-based method was employed, whereas for the NHP model, a luciferase transduction assay was utilized, as now clarified in **Supplementary Figure 26**. This discrepancy in methodology can significantly affect the interpretation of antibody titers and their impact on transduction efficiency.

9. Discussion: With the different AAV capsids used in the manuscript it would be helpful to have some discussion of the differences/similarities between these capsids, either within the discussion or possibly as an extension of supplementary table 1.

We thank the reviewer for this suggestion. We have extended **Supplementary Table 1**.

10. Discussion: Vector Mixture Competition: Within the mixture do the authors think there could be competition of vectors for uptake or a potential for antigenic competition with circulating AAV antibodies?

We hypothesize that at the doses used in our study, which are considerably lower (approximately 20,000-fold) than doses employed in the AAV5-based Roctavian™ marketed product, the system is unlikely to be saturated. Consequently, we don't expect significant competition among vectors for uptake. It's important to consider that receptor availability, vector affinity for those receptors, and the intrinsic transduction capabilities of each capsid can vary, and thus, influence whether competition occurs. As for antigenic competition with circulating AAV antibodies, it remains a theoretical possibility. The presence of a mixture of vectors could potentially alter the immune recognition and neutralization patterns compared to individual serotypes. Still, this is speculative and challenging to ascertain without direct, comparative evidence of single serotype administration versus a mixture in the presence of AAV antibodies. To fully understand the dynamics of vector competition and antigenic competition, additional targeted studies would be needed. We acknowledge that further research is needed to elucidate these phenomena fully.

Minor

1. Figure 1: Typo : "Dyalisis" – text box below the hepatic artery pump

This has now been corrected.

2. Figure 2b & 3b : The different AAR serotypes are quite difficult to distinguish – Consider reducing thickness of the data point outlines or adjust colours?

This has been adjusted as requested.

3. Discussion: Limitation : The system lacks circulating immune cells & other potential mechanisms of antibody mediated clearance

This point has been added into the discussion. It now reads (**Page 22**):

'While the liver explant system offers a valuable perspective for examining antibody neutralization in an environment containing liver-resident immune cells, it is limited by the exclusion of circulating immune cells and other systemic factors involved in antibody-mediated antigen clearance. These limitations may affect the system's ability to fully replicate the complex interactions and clearance mechanisms observed in the context of a whole-body.'

4. Methods: Murine xenografts: please provide details of volumes and flow rates – did these injections represent hydrodynamic delivery?

Volumes have now been added (150 μ l total volume) in the Methods section. The volumes used are far from the ~2mL used in hydrodynamic delivery. The materials and methods section now reads:

'Mice were randomly assigned to experiments and transduced via intravenous injection (lateral tail vein) with the indicated vector doses in a total volume of 150 μ l.'

5. Methods: Perfusate Media & Vectors: The perfusate media contains corticosteroids. There is some discussion as to whether treatment with corticosteroids could facilitate transduction, e.g., BMN 270-303 study in hemophilia A. Could this impact on uptake.

We have included a supplementary data exploring the possible selective pressures exerted by steroid use on AAV tropism and transgene expression (**Supplementary Figs 5-6**). Please also refer to Reviewer 1 – Answer 3 for further clarification.

6. Was any estimate made of full: empty capsids for the different constructs?

We did not estimate the ratio of full:empty capsids in our study. However, our vector purification process utilized a double cesium chloride gradient, a method well-regarded for its efficacy in selectively enriching for fully packaged capsids. We therefore hypothesize that our preparations are predominantly comprised of full capsids.

Reviewer #3 (Remarks to the Author):

Without a doubt, the concept of the study presented by Cabanes-Creus et al is milestone both for NMP Normothermic machine perfusion-based research as well as AAV development. It exemplifies the potential of this translational approach as it is closing a gap between stem cell and organoid based research, animal models and clinical trials.

The title is well chosen in the sense that the study indicates the potential of the methods applied in this study. However, while the study does successfully describe the feasibility of the concept it comes short with respect to the actual realization of the idea.

With respect to the sequence of experiments, the authors work backwards when they go from the work in human livers to the mouse and cell culture work. It might be advantageous to structure this in reverse order since the AAV based transduction in human livers would be the last rather than the first step.

We are thankful for Reviewer's recognition of our study as a milestone in both NMP normothermic machine perfusion-based research and AAV development. The acknowledgment of the potential of our translational approach in bridging the gap between various research modalities and clinical applications is highly encouraging. We concur that the study should be viewed as an initial proof of concept. It is our intention that this work serves as a foundation for subsequent publications that will build upon and expand the insights presented here. In line with the suggestions made, we have shifted the mouse and cell culture work to the supplementary materials. This reorganization allows us to focus on the human liver explant model, thereby emphasizing the novel aspects of our research and providing a more streamlined narrative. We appreciate the constructive feedback which has undoubtedly strengthened the presentation of our work.

The group has previously published their model for extended human liver preservation. In their paper (Nature Comm 2023:14:4755) they have described the model of prolonged full and split liver preservation with mixed results.

While the concept of this study is potentially groundbreaking, several aspects of the study represent significant limitations in the overall impact of the work. The key limitation is the limited number of organs included in the NMP work. Only two livers were used for these experiments. Considering the major inter-organ and perfusion variations, the findings need to be seen in the light of this limitation. One liver was excluded from transplantation due to biliary sepsis and the second liver stemmed from a DCD. Hence the impact of both the infection and the warm ischemia may be relevant and interfere with the findings.

We concur with the Reviewer's observations regarding the limitations due to the small number of organs used in the NMP work. We recognize the importance of considering the major inter-organ and perfusion variation in interpreting our findings. To address this, we have included a comprehensive discussion of these limitations in the discussion section of

our manuscript. This discussion emphasizes the need for caution in extrapolating the results and acknowledges the necessity for future studies to include a larger number of organs to validate and extend the findings of our pioneering work. It reads as follows:

'While we observed a notable correlation between the presence of neutralizing antibodies and the obstruction of liver uptake of specific capsids, we must acknowledge the potential impact of specific donor-related factors on our findings. Specifically, one of the livers was excluded from transplantation due to biliary sepsis, and the second liver was procured from a donor after circulatory death (DCD). Hence, the impact of both the infection and the warm ischemia may be relevant and could affect vector performance, adding a level of complexity to the interpretation of the results of our study. Furthermore, with only two livers utilized, our sample size is too small to account for the heterogeneity inherent in human liver physiology. Factors such as varying genetic backgrounds between both donors, as well as age, sex, and disease states, could potentially influence individual capsid performance. These variables underscore the significance of the limitation given the complex nature of liver functions and the variable response to normothermic machine perfusion (NMP) based on individual liver pathology and history. These constraints are critically highlighted in the context of our results and should be carefully considered.¹³ To achieve broader applicability and address these limitations, further research with a larger and more diverse range of liver organs is necessary to validate and generalize our findings.'

The continuous rise of AST in the perfusate indicates a progressive injury in addition to the I/R injury. This may be a limiting factor for the interpretation of the results. The split liver model, while attractive in terms of a case matched approach, has some significant limitations possible relevant in the context of the study. The two very different graft sizes and the diverse perfusion protocols and flows in the respective livers may influence the course of the experiment. The determination of the viability and function of livers as described in their previous study and as applied here has some limitations and defines a relatively liberal view on what can be considered viable and functional. The histological findings in the previous publication indicate coagulative necrosis between 0 and 100%.

For liver viability, we used criteria proposed by the VITTAL clinical trial, which include measurements like lactate levels below 2.5 mmol/L and other indicators such as bile production, pH, glucose metabolism, and sufficient hepatic arterial and portal vein flow. We continuously assessed the livers for hepatocellular viability using these criteria. As pointed out by the Reviewer, we also monitored liver biochemistry, including lactate clearance, which was maintained throughout perfusion until organ failure. We tracked liver function through bile production, maintaining an alkalotic pH indicative of preserved cholangiocyte function. Further assessments included levels of Factor V, oxygen consumption, which all remained stable or increased during perfusion until organ failure. As described in more detail in our recent publication (Nature Comm 2023:14:4755), we also looked at vascular hemodynamics, ensuring the independence of hepatic arterial and

portal venous pressure control while measuring vascular flow. Histopathology analysis showed that liver architecture was preserved during long-term perfusion, with low rates of coagulative necrosis or hepatocyte detachment.

We therefore, in this case, do not share the Reviewer's view of this being a 'relatively liberal view on what can be considered viable and functional'.

The different degrees of necrosis are also present in this study and seem to vary depending on the sampling site (supplementary fig 20). This inconsistency is also illustrated by the different AST and IL-6 levels in the perfusate.

In summary this all speaks towards the limitation arising from the small number of livers and the variables added through splitting.

We agree with the Reviewer, which is why these limitations are acknowledged and widely discussed throughout the manuscript.

The second liver received double the dose of AAV compared to the first liver. The limits the ability to compare the two experiments. A weight adjusted dosing seems more reasonable than such an approximation in the dosing.

Given the presence of anti-AAV neutralizing antibodies, it is our opinion that doubling the dose is equally as reasonable as maintaining the dose constant. We feel it is critical that the manuscript clearly outlines the study design and thus the fact that the dose was doubled, which will allow the readers to draw their own conclusions. With time, as more data accumulates and our understanding of the liver explant model improves, we will be able to look at the current data under a new light and we may be able to conclude then is doubling the dose had any influence on the final data or not.

The mechanisms involved in the anti-AAV neutralization remains speculative. The rapid vector removal is pointing in this direction, but further evidence for the relation is needed.

The nature of anti-AAV neutralization remains speculative as stated by the Reviewer. The text states (**Page 23**): '*...an effect we hypothesize to be mediated by liver-resident Kupffer cells*'.

I struggle to follow and understand the AAV transduction and transgene expression/function in the various models. It is difficult to understand the patterns and the mechanisms behind this. At current, this is rather confusing and not culminating to a coherent picture and understanding.

We have moved the detailed data on AAV transduction and transgene expression to the supplementary figures to streamline the main text. Additionally, for a clearer and more accessible overview, we have compiled and summarized the key results in **Supplementary Table 4**.

As the authors point out in the opening reference in their discussion, the study more

illustrates the potential of the methods applied rather than a fully conclusive assessment and implementation of these technologies. Several limitations are obvious and should be addressed now and in future studies. Nevertheless, the value and the visionary spirit of this research group deserves recognition.

We express our sincere gratitude to the Reviewer for their acknowledgment of the potential and visionary aspect of our research. We fully agree with the Reviewer's statement that our study is more illustrative of the potential of the methods applied than being a completely conclusive assessment. The limitations noted are indeed present and have been openly discussed in our manuscript. We are committed to addressing these limitations in ongoing and subsequent research efforts, and we consider this feedback instrumental in refining our methodologies and enhancing the applicability of our findings.

Minor:

Why were some barcoded constructs packaged in AAV2 and AAV-DJ underexpressed compared to mean transduction values? Why does this indicate successful transduction? (Suppl Fig 1).

We carried out the initial screen with n=25 barcodes to discard those that either caused overexpression or under expression of the transgene. The fact that barcodes can affect RNA stability and thus net transgene expression is well-known, which is why it is critical to validate each and every barcode prior to using it in a study. We have now complemented that study with additional studies using AAV-SYD12 *in vitro* and *in vivo*. (**Supplementary Fig. 3**).

The element of tropism in human livers is not addressed sufficiently. A more detailed analysis e.g. using confocal microscopy or single cell RNA analysis would help to achieve this. The mouse data are not conclusive and not immediately transferable to human livers. Again, an n=2 this might not be sufficient to conclusively demonstrate this.

We appreciate the Reviewer's suggestion for employing such techniques to address the element of tropism in human livers. While these methods are indeed valuable for in-depth analysis, we have relied on next-generation sequencing (NGS) and immunofluorescence to provide us with substantial information on the transduction profile in our study. NGS offers a high-throughput approach to evaluating tropism by quantifying vector genomes and transgene expression across the liver. Immunofluorescence complements this by allowing us to visualize transduction at the tissue level. Together, we believe these methods offer a robust overview of the AAV vectors' performance in the human liver. We acknowledge that there is inherent variability between organs, and that a sample size of two is a limitation that has been clearly stated within our discussion. Increasing the sample size in future studies would certainly enhance the conclusiveness of our findings. However, given the scope and resources of the current study, expanding on these methods was not feasible. We continue to believe that our current approach has provided valuable insights

into AAV transduction in human liver tissue and serves as a solid foundation for future research to build upon.

A figure showing histology of all livers at all time points would be helpful to compare the findings between livers/groups.

We thank the Reviewer for suggesting this. These data have now been added in main **Fig. 2**, and **Supplementary Figs 7 and 14**.

Reviewers' Comments:

Reviewer #1:

Remarks to the Author:

Comments on my previous concerns:

no more questions, but please make sure all images should meet the required quality.

I was also asked to comment on the concerns of Reviewer #2:

In general, the author has addressed the reviewer #2 concerns by providing additional information, justifications, and experimental data where necessary. However, some points need to be further emphasized.

The question #2 raised by reviewer#2 indicated the perfusion injury (ALT, IL6, cell death) could influence the AAV transduction, which would misguide the preclinical evaluation particularly in determining transduction efficacy. Reviewer #1 also expressed a similar concern in their previous query.

The author offered plausible explanation (due to Nabs rather than cell death) and HE staining evidence (sFig15, 7-10, 14, Fig 2d, 2h.) However, it is imperative to elaborate further on this concern.

Additionally, there appears to be missing text on page 11 as per the rebuttal letter, which should be addressed in the revised version.

We have now provided a clearer summary of which capsids contribute to this elevation.

The relevant text on Page 11 reads:

'We found an increase in the average vector copy number in the left graft (LLSG) that continued to be perfused with the AAV-containing perfusate following the organ split (Fig. 3c). In contrast, the vector copy number in the right graft, which received AAV-free perfusate after the organ split, remained relatively stable at the tested timepoints, ranging from 1.5 to 1 vg per haploid genome (Fig. 3c).'

missing text:

'The transduction data we obtained from the left graft (LLSG) appeared more variable than the data for the right graft (refer to top panel of Fig. 3d). We believe this was likely a consequence of vector recirculation in the perfusate following organ split, which facilitated ongoing transduction during the course of the study. Specifically, in the LLSG, we detected increased relative transduction of some variants over time (AAV-SYD12, AAV-LK03, and AAV-SEQ3), while others exhibited decreased relative transduction over time (AAV-FT01, AAV-hu.Lvr06, AAV-LK03-REDH). Transduction for the remaining vectors remained stable. Since these data present relative transduction, a decreasing contribution in the liver biopsies likely indicates either slower transduction kinetics or a decreasing amount of artificial signal stemming from the interstitial perfusate, rather than from vector uptake in hepatocytes. Indeed, the vectors showing a decreased relative transduction over time are the same one that stayed longer in the perfusate (Fig. 3c).'

Reviewer #3:

Remarks to the Author:

The authors have addressed all points of the review and provided informative feedback. Some of the limitations could be resolved, and the manuscript has been further improved. The remaining key limitations refer to the early and explorative character of the study. Hence the key question and decision remaining to be addressed is the small number of human liver experiments resulting in significant variations and limitations in the ability to draw strong conclusions. The liver splitting procedure is complicating the study. Ideally, a vector transfer experiment without liver splitting would be desirable to see the natural course of the vector transfer without the interference of the

additional procedure. I understand the ambition of testing the dynamic of the vector transfer further, but it is complicating the study.

Here are some minor comments:

The authors describe in the methods that "Prior to connecting the liver to the perfusion machine, we circulated the perfusate and activated the parallel dialysis circuit until the potassium, calcium and acidosis abnormalities in the stored blood were corrected"

How did the authors correct for potassium and calcium? This should be stated.

The cumulative bile volume amounts to approx. 770ml over the course of the perfusion. Has this volume be substituted? Also, after splitting the liver, was the same perfusate used as originally? Where FFPs added and where they matched with regard to the neutralizing ABs?

The authors have added the histopathologic findings of the two livers and state that "Histopathology analysis showed that liver architecture was preserved during long-term perfusion, with low rates of coagulative necrosis or hepatocyte detachment". This contrasts some of the data on coagulative necrosis and the respective pictures.

Liver 1 is deteriorating on day 5, liver 2 is deteriorating on day 8. This corresponds with histology of the ERG at hour 120, but confluent necrosis can also be seen in LLSG at 48 hours indicating early damage. The authors state that the experiment was terminate at one week, but they should mention that according to the criteria they are referring to (lactate 2.5 mmol/L) the damage occurs earlier than that. The AST levels after splitting are relatively high (9,762 U/L), indicating severe hepatocyte damage during and after splitting. Irrespective of the VITTAL criteria, the perfusion data show such a deterioration. While this is not the focus of the study, this should be mentioned.

REVIEWER COMMENTS

Reviewer #1 (Remarks to the Author):

Comments on my previous concerns: no more questions, but please make sure all images should meet the required quality.

- We thank the reviewer for their insightful suggestions and constructive feedback, which have greatly improved our manuscript. We have uploaded high resolution images (900dpi) in PDF format with this submission, and will provide the high-resolution TFF images upon final submission is that is the preferred format.

I was also asked to comment on the concerns of Reviewer #2:

In general, the author has addressed the reviewer #2 concerns by providing additional information, justifications, and experimental data where necessary. However, some points need to be further emphasized.

The question #2 raised by reviewer#2 indicated the perfusion injury (ALT, IL6, cell death) could influence the AAV transduction, which would misguide the preclinical evaluation particularly in determining transduction efficacy. Reviewer #1 also expressed a similar concern in their previous query.

The author offered plausible explanation (due to Nabs rather than cell death) and HE staining evidence (sFig15, 7-10, 14, Fig 2d, 2h.) However, it is imperative to elaborate further on this concern.

- We agree with both reviewers #1 and #2 that this is an important point that needs elaboration. We have further expanded the discussion on this matter, which can be found on page 23:

‘Thus, the observed elevation of IL-6 and ALT in donor 1 presents a potential confounding factor in the assessment of transduction efficacy of the capsids. Our analysis leads us to hypothesize that the primary determinant of the observed discrepancies between donor 1 and donor 2 is related to the presence of neutralizing antibodies in the case of donor 2. These antibodies likely exert a more pronounced effect on the transduction efficiency, overshadowing the potential impact of the elevated IL-6. Nonetheless, the role of IL-6 as a contributor to the variability in transduction cannot be entirely discounted and warrants further investigation.’

Additionally, there appears to be missing text on page 11 as per the rebuttal letter, which should be addressed in the revised version.

We have now provided a clearer summary of which capsids contribute to this elevation. The relevant text on Page 11 reads:

- The text below is present on page 11, middle of the second to last paragraph.

‘We found an increase in the average vector copy number in the left graft (LLSG) that continued to be perfused with the AAV-containing perfusate following the organ split (Fig. 3c). In contrast, the vector

copy number in the right graft, which received AAV-free perfusate after the organ split, remained relatively stable at the tested timepoints, ranging from 1.5 to 1 vg per haploid genome (Fig. 3c).'

missing text:

- We apologize for the misunderstanding. The text below is also present, although on pages 12/13, and not on page 11.

'The transduction data we obtained from the left graft (LLSG) appeared more variable than the data for the right graft (refer to top panel of Fig. 3d). We believe this was likely a consequence of vector recirculation in the perfusate following organ split, which facilitated ongoing transduction during the course of the study. Specifically, in the LLSG, we detected increased relative transduction of some variants over time (AAV-SYD12, AAV-LK03, and AAV-SEQ3), while others exhibited decreased relative transduction over time (AAV-FT01, AAV-hu.Lvr06, AAV-LK03-REDH). Transduction for the remaining vectors remained stable.

Since these data present relative transduction, a decreasing contribution in the liver biopsies likely indicates either slower transduction kinetics or a decreasing amount of artificial signal stemming from the interstitial perfusate, rather than from vector uptake in hepatocytes. Indeed, the vectors showing a decreased relative transduction over time are the same one that stayed longer in the perfusate (Fig. 3c).'

Reviewer #3 (Remarks to the Author):

The authors have addressed all points of the review and provided informative feedback. Some of the limitations could be resolved, and the manuscript has been further improved. The remaining key limitations refer to the early and explorative character of the study. Hence the key question and decision remaining to be addressed is the small number of human liver experiments resulting in significant variations and limitations in the ability to draw strong conclusions. The liver splitting procedure is complicating the study. Ideally, a vector transfer experiment without liver splitting would be desirable to see the natural course of the vector transfer without the interference of the additional procedure. I understand the ambition of testing the dynamic of the vector transfer further, but it is complicating the study.

- We sincerely appreciate the valuable comments provided by the reviewer. We agree with their insights and plan to incorporate them into the design and execution of our follow-up studies. Notably, we recognize the implications of splitting the livers in our experimental setup and, taking their advice into account, we intend to use whole livers in future research to avoid potential variability introduced by this procedure.

Here are some minor comments:

The authors describe in the methods that “Prior to connecting the liver to the perfusion machine, we circulated the perfusate and activated the parallel dialysis circuit until the potassium, calcium and acidosis abnormalities in the stored blood were corrected.” How did the authors correct for potassium and calcium? This should be stated.

- We have now added additional text in the Materials and Methods section on page 35:

‘Specifically, the parallel dialysis circuit used permits filtration of electrolytes across the dialysis filter. This allows the perfusate to be equilibrated with the dialysate to bring potassium, calcium (and other electrolytes) into the intended range.’

The cumulative bile volume amounts to approx. 770ml over the course of the perfusion. Has this volume be substituted?

- We did not substitute for the estimated bile volume. We routinely collect the bile and preserve it for further analyses with an overlaying layer of oil.

Also, after splitting the liver, was the same perfusate used as originally? Where FFPs added and where they matched with regard to the neutralizing ABs?

- We thank the reviewer for this comment. Yes, we used the same perfusate after the split. We have now clarified this in the Materials and Methods section (page 36), which now reads:

‘After the liver split, we ensured that the perfusate fresh frozen plasma (FFP) conditions were maintained consistent with the pre-split conditions.’

The authors have added the histopathologic findings of the two livers and state that “Histopathology analysis showed that liver architecture was preserved during long-term perfusion, with low rates of coagulative necrosis or hepatocyte detachment”. This contrasts some of the data on coagulative necrosis and the respective pictures. Liver 1 is deteriorating on day 5, liver 2 is deteriorating on day 8.

- We have now addressed this issue on Page 9 by adding the following text:

‘Overall, these results suggest that both livers were stable for a period of up to five and eight days, respectively.’

This corresponds with histology of the ERG at hour 120, but confluent necrosis can also be seen in LLSG at 48 hours indicating early damage. The authors state that the experiment was terminate at one week, but they should mention that according to the criteria they are referring to (lactate 2.5 mmol/L) the damage occurs earlier than that.

- We thank the reviewer for their insights. We have now incorporated this comment on Page 8, which now reads:

‘The degree of observed cell death was moderated and relatively constant across time (**Supplementary Figs 7-10**), although signs of confluent necrosis could be seen as early as 48 hours post-perfusion (**Supplementary Fig. 7**).’

The AST levels after splitting are relatively high (9,762 U/L), indicating severe hepatocyte damage during and after splitting. Irrespective of the VITTAL criteria, the perfusion data show such a deterioration. While this is not the focus of the study, this should be mentioned.

- We again thank the reviewer for this insight. We have clarified this point, as requested, on Pages 8/9, which now read:

‘The ALT levels peaked at 9,762 U L⁻¹ at 4 hours post-split for the LLSG, and gradually decreased to 1,663 U L⁻¹ on day 7.5 after splitting (**Supplementary Fig. 16**). This elevated ALT levels indicate severe hepatocyte damage during and after splitting.’